# LLDPE-like Polymers Accessible via Ethylene Homopolymerization Using Nitro-Appended 2-(Arylimino)pyridine-nickel Catalysts

Desalegn Demise Sage [1,2], Qiuyue Zhang [1], Ming Liu [1], Gregory A. Solan [1,3,*], Yang Sun [1] and Wen-Hua Sun [1,2,*]

[1] Key Laboratory of Engineering Plastics and Beijing National Laboratory for Molecular Sciences, Institute of Chemistry Chinese Academy of Sciences, Beijing 100190, China

[2] International School, University of Chinese Academy of Sciences, Beijing 100049, China

[3] Department of Chemistry, University of Leicester, University Road, Leicester LE1 7RH, UK

[*] Correspondence: gas8@leicester.ac.uk (G.A.S.); whsun@iccas.ac.cn (W.-H.S.); Tel.: +44-(0)116-2522096 (G.A.S.); +86-10-6255-7955 (W.-H.S.)

**Abstract:** Four examples of *para*-nitro substituted 2-(arylimino)pyridine-nickel(II) bromide complexes of general formula, $[2-\{(2,6\text{-R-}4\text{-NO}_2C_6H_2)N=CMe\}C_5H_4N]NiBr_2$, but differentiable by the steric/electronic properties displayed by the *ortho*-groups [R = *i*-Pr (**Ni1**), Et (**Ni2**), $CHPh_2$ (**Ni3**), $CH(4\text{-FPh})_2$ (**Ni4**)], have been prepared in good yield. For comparative purposes, the *meta*-nitro complex, $[2-\{(2,6\text{-}i\text{-}Pr_2\text{-}3\text{-NO}_2\text{-}4\text{-}(4\text{-FPh})_2C_6H)N=CMe\}C_5H_4N]NiBr_2$ (**Ni5**), has also been synthesized. The molecular structures of mononuclear **Ni3**·$xH_2O$ (x = 2, 3) and bromide-bridged dinuclear **Ni4** and **Ni5** are disclosed. Upon activation with either ethylaluminum dichloride ($EtAlCl_2$) or modified methylaluminoxane (MMAO), all precatalysts displayed good catalytic performance at operating temperatures between 30 °C and 60 °C with higher activities generally seen using $EtAlCl_2$ [up to $4.7 \times 10^6$ g PE (mol of Ni)$^{-1}$ h$^{-1}$]: **Ni2** ~ **Ni5** > **Ni1** ~ **Ni4** > **Ni3**. In terms of the resultant polyethylene (PE), **Ni4**/$EtAlCl_2$ formed the highest molecular weight of the series ($M_w$ up to $1.4 \times 10^5$ g mol$^{-1}$) with dispersities ($M_w/M_n$) ranging from narrow to broad ($M_w/M_n$ range: 2.2–24.4). Moreover, the melting temperatures ($T_m$) of the polymers generated via $EtAlCl_2$ activation fell in a narrow range, 117.8–126.0 °C, which resembles that seen for industrial-grade linear-low density polyethylene (LLDPE). Indeed, their $^{13}C$ NMR spectra revealed significant amounts of uniformly distributed long-chain branches (LCBs), while internal vinylene groups constituted the major type of chain unsaturation [vinylene:vinyl = 5.3:1 ($EtAlCl_2$) and 9.9:1 (MMAO)].

**Keywords:** nickel precatalysts; ethylene polymerization; *meta*-nitro vs. *para*-nitro group; thermal stability; medium molecular weight LLDPEs; controlled chain walking





## 1. Introduction

Over the years, the systematic development of Brookhart-type [1] nickel and palladium ethylene polymerization catalysts has garnered considerable attention on account of their exceptional ability to produce polyethylenes (PEs) with properties and applications that can meet the growing demand of day-to-day life [2–8]. With reference to nickel catalysts, the 2-(arylimino)pyridine family have shown a good track record for producing highly-branched PEs by using ethylene as the sole monomer in a single reactor system [7–9]. By comparison, their *N*,*N*-diaryl-*α*-diimine counterparts are able to generate various types of industrially important PEs ranging from ultra-high molecular weight (UHMWPE) through to LLDPE [10–12]. Elsewhere, *N*,*N*- and *N*,*O*-based nickel precatalysts have more recently been reported to demonstrate an aptitude to exhibit high activity and good thermal stability for ethylene homo-/copolymerization [13,14].

Our research team [15–17], and others [18–20], have made significant strides in the design of new *N,N*-nickel complexes that can produce α-olefins and a wide range of PE products. By reducing the steric properties of the *ortho*-substituents of the *N*-aryl groups, short-chain oligomers ($C_4$ and $C_6$) are predominantly produced [11,15,16,18–20]. On the other hand, the incorporation of electron withdrawing $NO_2$ groups into the ligand framework can not only improve catalytic activity but also enhance selectivity to produce PEs with vinyl/vinylene functionalities [11,21,22]. Conversely, by the introduction of more sterically bulky *N*-aryl substituents, or through modifications to the ligand backbone itself, chain transfer can be impeded resulting in solid PEs instead of oligomers [17,22–26]. In terms of controlling the branching architecture displayed by the PEs, targeted ligand design again plays a key role, which is aided by the reaction conditions employed (e.g., temperature, pressure, co-catalyst) [20–31].

With particular regard to the 2-(arylimino)pyridine ligand framework, a wide variety of nickel catalysts (**A**–**D**, Figure 1) [32–36], have been reported to generate moderate to highly branched value-added PEs. For example, **A** (Figure 1), incorporating sterically bulky benzhydryl (CHPh$_2$) units at the *ortho*- and *para*-positions of the *N*-aryl groups, are highly active [$>10^7$ g PE (mol Ni)$^{-1}$ h$^{-1}$] and have shown a capacity to polymerize ethylene to form branched PEs with narrow dispersity [32]. Furthermore, by increasing the steric hindrance at the imine carbon, a remarkable enhancement in the molecular weight and narrowing in the dispersity can be achieved [32]. Alternatively, by fusion of the iminopyridine ligand backbone with a cyclohexyl group to the central pyridine in **B** (Figure 1) and incorporation of *gem*-dimethyl groups at the C7 position, the thermal stability of the nickel precatalyst can be improved allowing access to highly branched PE [33]. In more recent studies, our group has reported the dibenzocycloheptyl-substituted **C** (Figure 1) [34], and phenyl-bridged bimetallic **D** (Figure 1) [35], and have shown these catalysts to promote low to high levels of branched PEs containing vinyl and vinylene unsaturation.

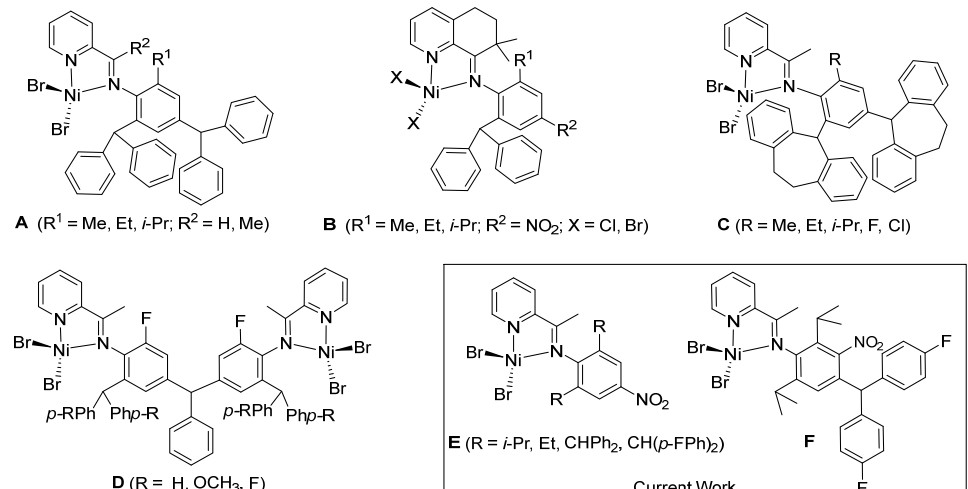

**Figure 1.** Developments in 2-(arylimino)pyridine-nickel(II) halide precatalysts (**A**–**F**) for ethylene polymerization.

However, one of the most common drawbacks encountered with 2-(arylimino) pyridine-containing nickel catalysts such as **A**–**D** (Figure 1) is their ready deactivation at elevated polymerization temperature due to limited steric protection of the active site imparted by the single *N*-aryl group. This leads to reduced catalytic activity and the production of PEs with undesirable properties (e.g., extremely low molecular weight, less control of branching and low melting temperatures). By contrast, *N,N*-diaryl-α-diimine-nickel and palladium precatalysts can promote ethylene polymerization under more industrially relevant conditions (operating temp.: 40–90 °C) and moreover produce PEs with a high selectivity for the branch distribution [37]. To the best of our knowledge, disclosures

of 2-iminopyridine-nickel catalysts that can operate at higher operating temperature are rare, while their use for the production of polymers resembling industrial grade medium molecular weight LLDPE has not been reported.

In this manuscript, we report a series of novel 2-(arylimino)pyridine-nickel(II) bromide ethylene polymerization precatalysts incorporating an electron-withdrawing nitro group at the *para*- (**E**) and *meta*-positions (**F**) of the *N*-aryl group (Figure 1). To explore the role of steric properties in **E**, the *ortho*-positions are systematically varied and include isopropyl, ethyl, benzhydryl and *para*-fluorinated benzhydryl, while in **F** the *ortho*-substitution is limited to isopropyl. A detailed ethylene polymerization study using **E** and **F** is performed that examines the effect of aluminum activator (type and amount), run temperature, run time and ethylene pressure. The intrinsic properties of the PEs (e.g., molecular weight, dispersity, microstructure) are thoroughly investigated and correlated with precatalyst structure, activator and physical conditions; comparisons with previous nickel catalysts are also highlighted. In addition, full synthetic and characterization details are presented for the free *N,N*-ligands and their complexes.

## 2. Results and Discussion

### 2.1. Synthesis and Characterization of *L1–L5* and Their Complexes *Ni1–Ni5*

The 2-(arylimino)pyridines, 2-{(2,6-R$_2$-4-NO$_2$C$_6$H$_2$)N=CMe}C$_5$H$_4$N [R = *i*-Pr (**L1**), Et (**L2**), CH(C$_6$H$_5$)$_2$ (**L3**), CH(4-F-C$_6$H$_4$)$_2$ (**L4**)] and 2-{(2,6-*i*-Pr$_2$-3-NO$_2$-4-(4-F-C$_6$H$_4$)$_2$C$_6$H) N=CMe}C$_5$H$_4$N (**L5**), have been prepared in good yield by the acid-catalyzed condensation reaction of 2-acetylpyridine with the corresponding aniline (**A1–A5**) (Scheme 1) [38,39]. Anilines, 2,6-R$_2$-4-NO$_2$C$_6$H$_2$NH$_2$ (R = *i*-Pr **A1**, Et **A2**) are commercially available, while 2,6-R$_2$-4-NO$_2$C$_6$H$_2$NH$_2$ (R = CH(C$_6$H$_5$)$_2$ **A3**, CH(4-F-C$_6$H$_4$)$_2$ **A4**) along with (2,6-*i*-Pr$_2$-3-NO$_2$-4-(4-F-C$_6$H$_4$)$_2$C$_6$HNH$_2$) (**A5**) can be prepared in good yield by zinc (II) chloride mediated Friedel–Crafts alkylation reactions (see Schemes S1 and S2). All new organic compounds have been characterized by $^1$H NMR, $^{13}$C NMR, FT-IR spectroscopy and elemental analysis.

| | Ni1 | Ni2 | Ni3 | Ni4 |
|---|---|---|---|---|
| R | *i*-Pr | Et | CHPh$_2$ | CH(4-FPh)$_2$ |
| Yield (%) | 91 | 91 | 56 | 80 |

**Scheme 1.** Synthetic route to nitro-containing **L1–L5** and their corresponding 2-iminopyridine-nickel(II) bromide complexes, **Ni1–Ni5**.

Interaction of **L1–L5** with (DME)NiBr$_2$ (DME = 1,2-dimethoxyethane) in a mixture of dichloromethane and ethanol (1:1 v/v) afforded on work-up the nickel(II) bromide complexes, [2-{(2,6-R$_2$-4-NO$_2$C$_6$H$_2$)N=CMe}C$_5$H$_4$N]NiBr$_2$ [R = *i*-Pr (**Ni1**), Et (**Ni2**), CHPh$_2$ (**Ni3**), CH(4-FPh)$_2$ (**Ni4**)] and [2-{(2,6-*i*-Pr$_2$-3-NO$_2$-4-(4-F-C$_6$H$_5$)$_2$C$_6$H)N=CMe}C$_5$H$_4$N]NiBr$_2$ (**Ni5**), in good yield (Scheme 1). All nickel complexes were air-stable, green or brown powders which have been fully characterized by FT-IR spectroscopy, elemental analysis and in the case of **Ni3**, **Ni4** and **Ni5** by single-crystal X-ray diffraction.

Single crystals of **Ni3**, **Ni4** and **Ni5** suitable for the X-ray determinations were grown at ambient temperature by diffusing either diethyl ether into a dichloromethane solution of **Ni3** or hexane into dichloromethane solutions of **Ni4** and **Ni5**. For **Ni3**, inspection of the unit cell (Figure S1) indicated that adventitious coordination of water had occurred during crystallization to form **Ni3**·$xH_2O$ (x = 2, 3). Furthermore, closer examination of the unit cell revealed the presence of three independent molecules, one based on a neutral species, **L3**$NiBr_2(OH_2)_2$ (molecule A, x = 2) and the two others based on a cationic complex [**L3**$NiBr(OH_2)_3$]Br (molecules B and C, x = 3). Only molecules A and B will be discussed in any detail as cationic B and C show minimal differences. Views of A and B are depicted in Figure 2a,b, while selected bond lengths and angles are given in Table 1. For **Ni4** and **Ni5**, the corresponding bond parameters can be seen in Table 2, while representations of their structures are shown in Figures 3 and 4.

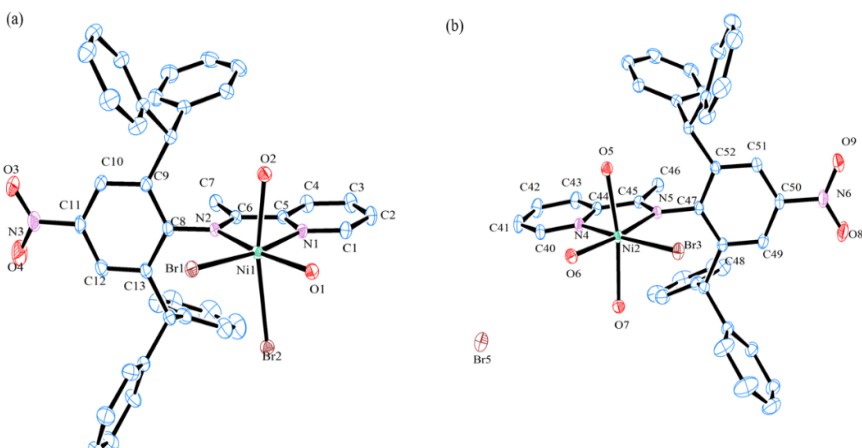

**Figure 2.** Co-crystallized structures present in **Ni3**·$xOH_2$ (x = 2, 3): (**a**) neutral (**L3**)$NiBr_2(OH_2)_2$ (molecule A) and (**b**) cationic [(**L3**)$NiBr_2(OH_2)_3$]Br (molecule B). Thermal ellipsoids are shown at the 30% probability level, and hydrogen atoms are omitted for clarity.

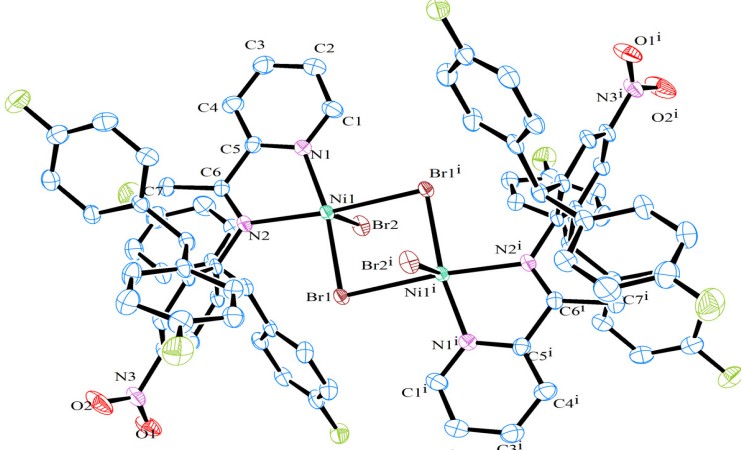

**Figure 3.** Molecular structure of **Ni4**. Thermal ellipsoids are shown at the 30% probability level, and hydrogen atoms are omitted for clarity. The atoms denoted with 'i' have been generated by symmetry.

**Table 1.** Selected bond lengths (Å) and angles (°) in **Ni3**·xOH$_2$ (x = 2, 3).

| (L3)NiBr$_2$(H$_2$O)$_2$ (Molecule A) | | [(L3)NiBr(H$_2$O)$_3$]Br (Molecule B) | |
|---|---|---|---|
| **Bond lengths (Å)** | | | |
| Ni(1)–N(1) | 2.065(2) | Ni(2)–N(4) | 2.043(2) |
| Ni(1)–N(2) | 2.149(2) | Ni(2)–N(5) | 2.123(2) |
| Ni(1)-Br(1) | 2.5280(5) | Ni(2)-Br(3) | 2.5290(5) |
| Ni(1)–O(1) | 2.1298(19) | Ni(2)–O(5) | 2.1004(19) |
| Ni(1)–Br(2) | 2.5634(5) | Ni(2)–O(6) | 2.055(2) |
| Ni(1)–O(2) | 2.139(2) | Ni(2)–O(7) | 2.113(3) |
| N(2)–C(6) | 1.291(3) | N(5)–C(45) | 1.293(3) |
| N(1)–C(5) | 1.347(3) | N(4)–C(44) | 1.351(3) |
| N(3)–O(3) | 1.225(4) | N(6)–O(8) | 1.219(4) |
| N(3)–O(4) | 1.217(4) | N(6)–O(9) | 1.223(4) |
| **Bond angles (°)** | | | |
| N(1)–Ni(1)–N(2) | 78.08(8) | N(4)–Ni(2)–N(5) | 78.95(8) |
| N(1)–Ni(1)–Br(1) | 171.24(6) | N(4)–Ni(2)–Br(3) | 170.45(6) |
| N(1)–Ni(1)–Br(2) | 92.21(6) | N(4)-Ni(2)–O(5) | 90.86(8) |
| N(1)–Ni(1)–O(1) | 92.30(8) | N(4)–Ni(1)–O(6) | 93.78(9) |
| N(1)–Ni(1)–O(2) | 86.37(9) | N(4)–Ni(1)–O(7) | 87.96(9) |
| N(2)–Ni(1)–Br(1) | 94.40(6) | N(5)–Ni(2)–Br(3) | 91.83(6) |
| N(2)–Ni(1)–Br(2) | 100.33(6) | N(5)–Ni(2)–O(6) | 172.00(8) |
| N(2)–Ni(1)–O(1) | 170.34(8) | N(5)–Ni(2)–O(7) | 96.92(8) |
| O(1)–Ni(1)–Br(1) | 95.11(6) | O(6)–Ni(2)–Br(3) | 95.57(8) |
| O(1)–Ni(1)–Br(2) | 80.81(6) | O(6)–Ni(2)–O(5) | 83.01(8) |
| O(1)–Ni(1)-O(2) | 84.72(8) | O(6)–Ni(2)–O(7) | 85.05(8) |
| O(2)–Ni(1)–Br(1) | 89.65(6) | O(7)–Ni(2)–Br(3) | 90.75(6) |
| O(2)–Ni(1)–Br(2) | 165.39(6) | O(7)–Ni(2)–O(5) | 168.90(8) |
| Br(1)–Ni(1)–Br(2) | 93.583(16) | Br(3)–Ni(2)–O(5) | 92.19(6) |
| O(3)–N(3)–O(4) | 123.9(3) | O(8)–N(6)–O(9) | 123.2(3) |
| C(6)–N(2)–C(8) | 118.4(2) | C(45)–N(5)–C(47) | 120.9(2) |

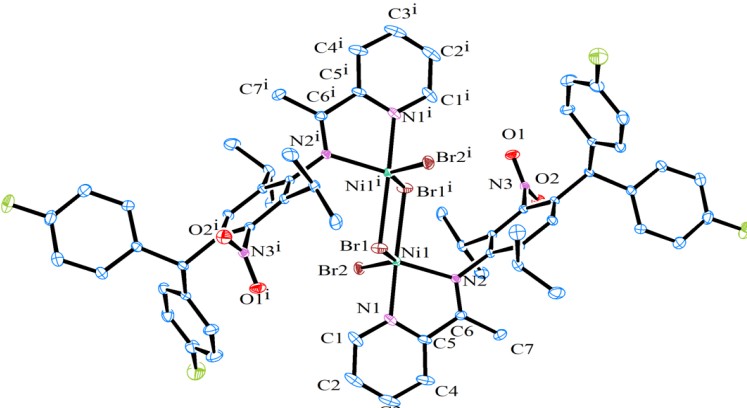

**Figure 4.** Molecular structure of **Ni5**. Thermal ellipsoids are shown at the 30% probability level, and hydrogen atoms are omitted for clarity. The atoms denoted with 'i' have been generated by symmetry.

**Table 2.** Selected bond lengths (Å) and angles (°) for **Ni4** and **Ni5**.

| | Ni4 | Ni5 |
|---|---|---|
| **Bond lengths (Å)** | | |
| Ni(1)–N(1) | 2.037(4) | 2.048(2) |
| Ni(1)–N(2) | 2.151(3) | 2.0518(17) |
| Br(1)–Ni(1) | 2.5100(8) | 2.4863(4) |
| Br(1)–Ni(1$^i$) | 2.6049(7) | 2.5562(3) |
| Br(2)–Ni(1) | 2.5573(8) | 2.4167(4) |
| N(2)–C(8) | 1.442(5) | 1.445(2) |
| N(2)–C(6) | 1.283(3) | 1.287(3) |
| N(1)–C(5) | 1.353(6) | 1.344(3) |
| **Bond angles (°)** | | |
| Ni(1)–Br(1)–Ni(1$^i$) | 87.52(3) | 94.984(14) |
| Br(1)–Ni(1)–Br(1$^i$) | 92.48(3) | 85.017 (14) |
| Br(1)–Ni(1)–Br(2) | 94.08(3) | 138.298(17) |
| Br(2)–Ni(1)–Br(1$^i$) | 85.35(2) | 95.917(2) |
| N(2)–Ni(1)–Br(1) | 92.88(10) | 116.3(5) |
| N(2)–Ni(1)–Br(1$^i$) | 173.89(10) | 100.31(5) |
| N(2)–Ni(1)–Br(2) | 97.19(9) | 104.58(5) |
| N(1)–Ni(1)–Br(1) | 170.52(11) | 91.13 (6) |
| N(1)–Ni(1)–Br(1$^i$) | 95.90(10) | 175.47 (6) |
| N(1)–Ni(1)–Br(2) | 91.01(11) | 88.55(6) |
| N(1)–Ni(1)–N(2) | 78.54(14) | 79.25 (7) |
| C(6)–N(2)–Ni(1) | 113.9(3) | 115.09(14) |
| C(8)–N(2)–Ni(1) | 125.9(3) | 124.87(13) |

[1] The elements denoted with 'i' have been generated by symmetry.

For **Ni3**·xH$_2$O (x = 2, 3), both molecules A and B comprise a single metal center that is surrounded by two nitrogen atoms belonging to the chelating 2-(2,6-dibenzhydryl-4-nitrophenylimino)pyridine ligand. In A, two terminal bromides [Br(1) and Br(2)] and two water ligands [O(1) and O(2)] complete the coordination sphere of the six-coordinate neutral species. By contrast in cationic B, three water molecules [O(5), O(6) and O(7)] and a bromide [Br(3)] fill the remaining four coordination sites, while the non-coordinating bromide anion, Br(5), charge balances the complex. In both molecules, the geometry about nickel can be best described as distorted octahedral with the equatorial belt in each case based on N/N/Br/O donors but differing in the composition of the axial sites, O/Br (molecule A) and O/O (molecule B). Comparison of the corresponding cis angles highlights the distortions apparent in each molecule [A: Br(1)–Ni(1)–Br(2) 93.583(16)°, O(1)–Ni(1)–O(2) 84.72(8)°, O(2)–Ni(1)–Br(1) 89.65(6)° vs. B: Br(3)–Ni(2)–O(5) 92.19(6)°, O(6)–Ni(2)–O(7) 85.05(8)°, O(7)–Ni(2)–Br(3) 90.75(6)°]; similar distortions can be observed within previously reported mononuclear structures of this class of complex [32,34,35,37]. In terms of the Ni-N distances, the Ni-N$_{pyridine}$ length is significantly shorter than the Ni–N$_{imine}$ [A: 2.065(2) Å vs. 2.149(2) Å for A and B: 2.043(2) Å vs. 2.123(2) Å], highlighting the improved donor properties of the former; similar observations have been noted elsewhere [32,34]. The bond parameters involving the *para*-nitro group are not exceptional and are in line with the expected sp$^2$-hybridization. In addition, there are a number of inter- and intramolecular hydrogen bonding interactions involving the aqua ligands including OH···Br contacts (Figure S1).

By contrast, **Ni4** and **Ni5** adopt binuclear structures, in which each nickel center is bridged by two bromide ligands and further bound by an *N,N*-chelating 2-(arylimino)pyridine ligand (aryl = 2,6-bis(4,4-difluorobenzhydryl)-4-nitrophenyl **Ni4**, 2,6-diisopropyl-3-nitro-4-(4,4-difluorobenzhydryl)phenyl **Ni5**) and a terminal bromide. Despite both being five-coordinate, their geometries are best described as either distorted square-based pyramidal for **Ni4** (τ value = 0.06) or trigonal bipyramidal for **Ni5** (τ value = 0.99) [32]. The origin of this structural variation is uncertain, but it likely derives from the differences in the steric properties imposed

by the *ortho*-substituents (*viz.* 4,4-difluorobenzhydryl **Ni4**, isopropyl **Ni5**). As with **Ni3**·$x$H$_2$O ($x$ = 2, 3), the Ni-N$_{pyridine}$ bond lengths in both complexes (**Ni4** and **Ni5**) are shorter than corresponding Ni–N$_{imine}$ distances [2.037(4) Å vs. 2.151(3) Å for **Ni4** and 2.048(2) Å vs. 2.0518(17) Å for **Ni5**] once again reflecting the good donor properties of the pyridine. Inspection of the literature reveals that the (*N,N*)NiBr($\mu$-Br$_2$)NiBr(*N,N*) structural motif is commonplace for 2-(arylimino)pyridine-nickel(II) halide complexes [32]. There are no intermolecular contacts of note in these two cases.

The FT-IR spectra of **Ni1–Ni5** displayed $v$(C=N) absorption bands between 1626 and 1598 cm$^{-1}$, which are lessened when compared to 1652 and 1643 cm$^{-1}$ in their free ligands, **L1–L5**. This reduction in wavenumber provides support for the effective coordination between the metal center and the nitrogen atom of the imine group [26]. Additionally, the microanalytical data for the nickel complexes were fully consistent with compositions of the type, **LNiBr$_2$**.

### 2.2. Catalytic Evaluation for Ethylene Polymerization

#### 2.2.1. Co-Catalyst Screening

To pinpoint the most compatible co-catalyst to promote effective ethylene polymerization, two aluminoxanes and three alkyl aluminum chlorides were initially evaluated with **Ni4** employed as the test precatalyst: methylaluminoxane (MAO), modified methylaluminoxane (MMAO: containing 20–25% Al(*i*-Bu)$_3$), ethylaluminum sesquichloride (Et$_3$Al$_2$Cl$_3$, EASC), diethyl aluminum chloride (Et$_2$AlCl) and ethylaluminum dichloride (EtAlCl$_2$). All runs were performed at 30 °C in toluene under 10 atm of ethylene pressure with the run time maintained at 30 min (Table 3). All co-catalysts successfully demonstrated an aptitude for producing active species from **Ni4** with EtAlCl$_2$ being the standout performer (activity = 3.8 × 10$^6$ g PE (mol Ni)$^{-1}$ h$^{-1}$). In terms of the relative performance, the activity with respect to co-catalyst fell in the following order: EtAlCl$_2$ > MMAO > MAO > EASC > Et$_2$AlCl. As a result, EtAlCl$_2$ and MMAO were selected to further optimize the polymerization parameters using **Ni4**.

**Table 3.** Ethylene polymerization using **Ni4** with a range of different aluminum co-catalysts [a].

| Run | Co-Cat. | Al:Ni | Activity [b] | $M_w$ [c] | $M_w/M_n$ [c] | $T_m$ (°C) [d] |
|-----|---------|-------|-------------|-----------|---------------|----------------|
| 1 | MAO | 2000 | 1.9 | 78.9 | 21.6 | 67.8, 121.4 |
| 2 | MMAO | 2000 | 2.1 | 7.2 | 3.4 | 93.4, 110.6 |
| 3 | EASC | 600 | 1.8 | 130.4 | 28.5 | 118.8 |
| 4 | Et$_2$AlCl | 600 | 0.2 | 63.2 | 14.5 | 125.3 |
| 5 | EtAlCl$_2$ | 600 | 3.8 | 61.5 | 12.9 | 125.9 |

[a] Conditions: 2.0 μmol of **Ni4**, 100 mL of toluene, 10 atm C$_2$H$_4$, 30 °C, 30 min; [b] Values in units of 10$^6$ g (PE) (mol Ni)$^{-1}$ h$^{-1}$; [c] $M_w$: kg mol$^{-1}$, determined by GPC; [d] Determined by DSC.

#### 2.2.2. Ethylene Polymerization Using **Ni1–Ni5** in the Presence of EtAlCl$_2$

To identify an optimum set of reaction conditions using **Ni4**/EtAlCl$_2$, the effect of adjusting the run temperature, Al:Ni molar ratio and run time have all been studied with the pressure initially maintained at P$_{C2H4}$ = 10 atm; the results of this evaluation are gathered in Table 4.

Firstly, the influence of modifying the Al:Ni molar ratio on the performance of **Ni4**/EtAlCl$_2$ was investigated with the run temperature fixed at 30 °C and the run time at 30 min. By increasing the molar ratio from 400:1 to 800:1 (runs 1–5, Table 4), the maximum catalytic activity of 3.8 × 10$^6$ g PE (mol of Ni)$^{-1}$ h$^{-1}$ was achieved at 600:1 (run 3, Table 4). Further increasing the amount of EtAlCl$_2$ resulted in a gradual decrease in the activity in a manner similar to that seen in a number of previous reports [38–42].

**Table 4.** Optimization of the polymerization conditions using **Ni4**/EtAlCl$_2$ and catalytic evaluation of **Ni1–Ni3** and **Ni5** [a].

| Run | Precat. | T (°C) | t (min) | Al:Ni | Activity [b] | $M_w$ [c] | $M_w/M_n$ [c] | $R_i$ [d] | $R_t$ [e] | $T_m$ (°C) [f] |
|-----|---------|--------|---------|-------|--------------|-----------|---------------|-----------|-----------|----------------|
| 1 | **Ni4** | 30 | 30 | 400 | 0.9 | 129.5 | 17.7 | 64.8 | 248.4 | 119.3 |
| 2 | **Ni4** | 30 | 30 | 500 | 2.8 | 137.7 | 14.2 | 197.5 | 570.0 | 126.0 |
| 3 | **Ni4** | 30 | 30 | 600 | 3.8 | 61.5 | 12.9 | 268.8 | 1576.1 | 125.9 |
| 4 | **Ni4** | 30 | 30 | 700 | 3.2 | 61.0 | 7.6 | 228.2 | 796.0 | 123.3 |
| 5 | **Ni4** | 30 | 30 | 800 | 3.1 | 57.1 | 8.3 | 218.9 | 895.0 | 122.4 |
| 6 | **Ni4** | 20 | 30 | 600 | 3.1 | 63.3 | 9.9 | 223.9 | 982.2 | 121.0 |
| 7 | **Ni4** | 40 | 30 | 600 | 3.5 | 45.1 | 16.8 | 252.4 | 2632.9 | 124.1 |
| 8 | **Ni4** | 50 | 30 | 600 | 3.4 | 32.9 | 18.2 | 245.3 | 3822.2 | 123.7 |
| 9 | **Ni4** | 60 | 30 | 600 | 3.0 | 30.9 | 20.5 | 213.2 | 3978.1 | 119.3 |
| 10 | **Ni4** | 30 | 10 | 600 | 2.7 | 53.0 | 2.2 | 190.8 | 2247.2 | 124.5 |
| 11 | **Ni4** | 30 | 20 | 600 | 2.9 | 56.3 | 6.5 | 205.6 | 671.3 | 121.9 |
| 12 | **Ni4** | 30 | 40 | 600 | 3.7 | 67.3 | 13.3 | 259.1 | 1435.9 | 124.5 |
| 13 | **Ni4** | 30 | 50 | 600 | 3.5 | 69.6 | 12.9 | 252.6 | 1316.5 | 124.8 |
| 14 | **Ni4** | 30 | 60 | 600 | 3.1 | 81.6 | 24.4 | 223.9 | 1878.0 | 124.6 |
| 15 [g] | **Ni4** | 30 | 30 | 600 | 2.1 | 32.2 | 5.6 | 149.0 | 727.0 | 124.0 |
| 16 [h] | **Ni4** | 30 | 30 | 600 | - | - | - | - | - | - |
| 17 | **Ni1** | 30 | 30 | 600 | 3.1 | 13.1 | 12.7 | 220.3 | 5965.2 | 118.6 |
| 18 | **Ni2** | 30 | 30 | 600 | 4.7 | 22.5 | 7.9 | 337.2 | 3331.0 | 117.8 |
| 19 | **Ni3** | 30 | 30 | 600 | 2.6 | 7.4 | 8.9 | 188.2 | 6346.2 | 123.2 |
| 20 | **Ni5** | 30 | 30 | 600 | 4.6 | 23.3 | 22.2 | 329.4 | 8833.7 | 120.3 |

[a] Conditions: 2.0 μmol of nickel precatalyst, 100 mL toluene, and 10 atm of C$_2$H$_4$; [b] $10^6$ g of PE (mol of Ni)$^{-1}$ h$^{-1}$; [c] $M_w$: kg mol$^{-1}$, determined by GPC; [d] Rate of monomer insertion, in units of mmol/h; [e] Rate of chain termination, in units of μmol/h, [f] Determined by DSC; [g] 5 atm of C$_2$H$_4$; [h] 1 atm of C$_2$H$_4$.

With regard to the polymer, variations in the molar ratios over the aforementioned range led to the molecular weight of the PE displaying a wide variety in values (57.1–137.7 kg mol$^{-1}$). In particular, the molecular weight was at its highest when using the lowest molar ratio (runs 1 and 2, Table 4), and indeed twice as high as the molecular weights obtained using larger amounts of EtAlCl$_2$ (runs 3–5, Table 4). Again, related trends have been observed for structurally related nickel precatalysts [23]. To explain this finding, it would appear that at a higher concentration, the aluminum species interrupts the chain propagation and enhances the termination process (e.g., by chain transfer to aluminum). Strong agreement for the latter is provided by the increased value of the rate of chain termination ($R_t$ from 796.0 to 895.0 μmol/h) and the decreased value of the rate of monomer insertion ($R_i$ from 228.2 to 218.9 mmol/h) (runs 4 and 5, Table 4) [43,44]. A similar conclusion has been reached for *ortho*-unsymmetrically substituted 2-(arylimino)pyridine [35], and bis(imino)pyridine nickel precatalyts [42,45–47]. Moreover, all PEs possessed broad dispersities ($M_w/M_n$ range: 7.6–17.7), which is borne out by their GPC traces suggesting the catalyst displayed multi-site characteristics (Figure 5).

Secondly, to explore the thermal stability of **Ni4**/EtAlCl$_2$, the polymerization runs were conducted at a series of fixed temperatures with the Al:Ni molar ratio kept at 600:1 and the reaction time at 30 min (runs 3 and 6–9, Table 4). On raising the temperature incrementally from 20 °C to 60 °C, a peak activity of $3.8 \times 10^6$ g PE (mol of Ni)$^{-1}$ h$^{-1}$ was achieved at 30 °C (run 3, Table 4). Above this temperature, the activity gently declined reaching a level of $3.0 \times 10^6$ g PE (mol of Ni)$^{-1}$ h$^{-1}$ when the temperature reached 60 °C (run 9, Table 4). Given this modest drop in catalytic activity (*ca.* 20%), **Ni4**/EtAlCl$_2$ can be considered as showing appreciable thermal stability and indeed offers potential to be operated at more industrially relevant temperatures (e.g., 70–115 °C) [12–14,46]. In terms of the polymers, their molecular weights decreased progressively from 63.3 kg mol$^{-1}$ at 20 °C to 30.9 kg mol$^{-1}$ at 60 °C (runs 3 and 6–9, Table 4), which can be mainly credited to the sharply increased chain termination rate ($R_t$ range: 982.2 to 3978.1 μmol/h) on increasing the run temperature from 20 °C to 60 °C (runs 3 and 6–9, Table 4). Similar conclusions have been drawn elsewhere [43,44]. Additionally, partial deactivation of the active species, and lower ethylene solubility in

toluene at higher operating temperature could be other contributing factors. By contrast, the polymer dispersity ($M_w/M_n$) considerably broadened from 9.9–20.5 and shifted to lower molecular weight with rising temperature (Figure 6). Nonetheless, the melting temperatures ($T_m$) of the PEs remained high and fell in a relatively narrow range ($T_m$ = 119.3–125.9 °C), which is consistent with limited change in the microstructure of these polymeric materials over the range in run temperatures. It is worthy of note that it is rare to achieve such thermally Table 3 iminopyridine-nickel complexes and to produce polyethylene with comparable molecular weights at such elevated polymerization temperatures.

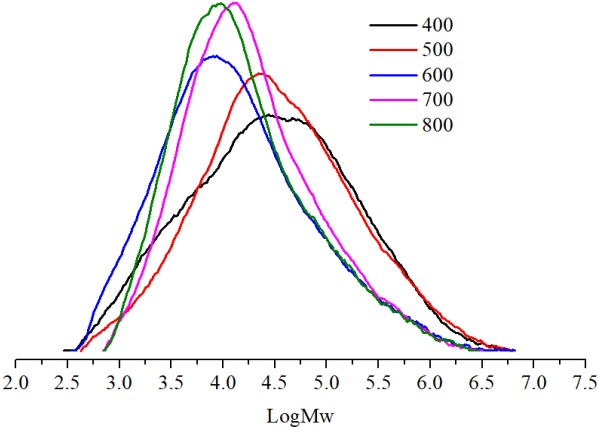

**Figure 5.** GPC curves of the polyethylene produced using **Ni4**/EtAlCl₂ at different molar ratios (runs 1–5, Table 4).

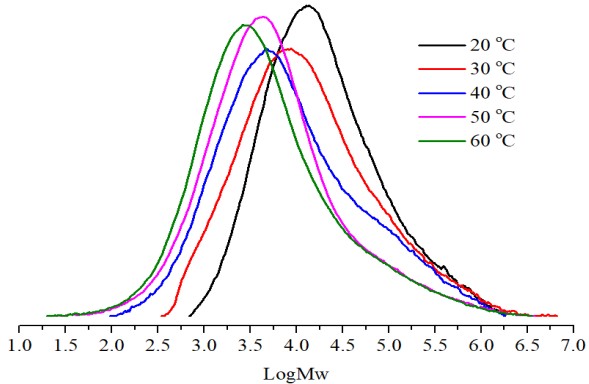

**Figure 6.** GPC curves of the polyethylene produced using **Ni4**/EtAlCl₂ at different run temperatures (runs 3 and 6–9, Table 4).

Thirdly, to probe the effectiveness of **Ni4**/EtAlCl₂ over the course of time, we carried out separate polymerization runs at 10, 20, 30, 40, 50 and 60 min (runs 3 and 10–14, Table 4). The peak activity of $3.8 \times 10^6$ g PE (mol of Ni)$^{-1}$ h$^{-1}$ was achieved over 30 min suggesting a lengthy induction period (run 3, Table 4). Noticeably, on extending the reaction time to 60 min the catalytic activity only slightly reduced to $3.1 \times 10^6$ g PE Ni mol$^{-1}$ h$^{-1}$ (run 14, Table 4), highlighting the appreciable lifetime of the active species. Such a modest lowering of catalytic activity over prolonged run time would suggest deactivation is kept to a minimum, indeed a similar conclusion was recently reached for *N,N*-diaryl-$\alpha$-diimine-based nickel catalysts [40]. Perhaps more importantly, when compared with structurally related 2-(arylimino)pyridine nickel catalysts, these performance characteristics demonstrate a key improvement as catalytic activity can sometimes be reduced more dramatically over similar extended run times [34]. In terms of molecular weight, all the polymers displayed high values ($M_w$ range: 53.0–81.6 kg mol$^{-1}$) that consistently increased as the run time elapsed. Remarkably, the dispersity of the polymer showed it to be more

sensitive to reaction time than other polymerization parameters with its $M_w/M_n$ values rapidly increasing from 2.2 to 24.4 over the duration. As is evident from the GPC traces, the broadness of the unimodal peaks increased and shifted towards the high molecular weight region upon prolonging the run time to 50 min (Figure 7). At 60 min, multimodality was evident resulting from the formation of additional lower molecular weight components which is demonstrated by the appearance of double shoulder peaks.

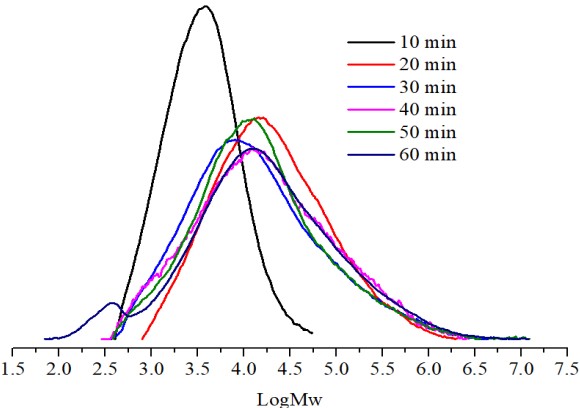

**Figure 7.** GPC curves of the polyethylene produced using **Ni4**/EtAlCl₂ at different reaction times (runs 3 and 10–14, Table 4).

To complement the runs performed using **Ni4**/EtAlCl₂ at $P_{C2H5}$ = 10 atm, a series of tests were additionally conducted under 1 atm and 5 atm ethylene pressure (runs 3, 15 and 16, Table 4). Polymerizations at 5 atm resulted in almost half the activity and molecular weight of the PE obtained at 10 atm (runs 3 vs. 15, Table 4). By contrast at 1 atm, only trace amounts of PE could be isolated (run 16, Table 4), highlighting a critical amount of ethylene pressure needed to ensure significant polymer formation.

From the above, the optimum set of conditions using **Ni4**/EtAlCl₂ can be summarized as $P_{C2H4}$ = 10 atm, run temperature = 30 °C, run time = 30 min and Al:Ni molar ratio = 600:1. Based on these reaction parameters, the remaining nickel precatalysts, **Ni1**–**Ni3** and **Ni5**, were then investigated; the results of these tests are tabulated in Table 4 (runs 3 and 17–20). In terms of relative performance the order follows: **Ni2** [2,6-di(Et)]~**Ni5** [2,6-di(*i*-Pr)-3-NO₂] > **Ni4** [2,6-di(CH(4-F-C₆H₄)₂] > **Ni1** [2,6-di(*i*-Pr)] > **Ni3** [2,6-di(CH(C₆H₅)₂)]. In general, the polymerization activity increased on decreasing the steric hindrance exerted by the *ortho*-substituents and by introducing electron-withdrawing substituents (e.g., F). For example, replacing Et with *i*-Pr at the *ortho*-positions decreased the catalytic activity nearly by half ($4.7 \times 10^6$ g PE (mol of Ni)$^{-1}$ h$^{-1}$ vs. $2.6 \times 10^6$ g PE (mol of Ni)$^{-1}$ h$^{-1}$). The lowest catalytic activity was displayed by **Ni3** while its fluorinated counterpart **Ni4** was noticeably more active. Of particular note, 2,6-diisopropyl-containing **Ni5** bearing remote *meta*-NO₂ and *para*-4,4-difluorobenzydryl groups showed the second-highest catalytic activity [$4.6 \times 10^6$ g (PE mol of Ni)$^{-1}$ h$^{-1}$] which was indeed greater than its *para*-nitro comparator **Ni1**. A similar comparison was made previously, but in that case the *meta*-NO₂ nickel catalyst exhibited the lowest catalytic activity of the systems explored [23].

In terms of polymer molecular weight, that formed by **Ni4**/EtAlCl₂ showed the highest of this series (61.5 kg mol$^{-1}$), while **Ni5**/EtAlCl₂ the second highest (23.3 kg mol$^{-1}$). Indeed, these $M_w$ values were much higher than those mentioned in previous studies [35], highlighting the positive role of electron-withdrawing groups on chain propagation. By contrast, **Ni3**, the non-fluorinated analogue of **Ni4**, gave polymers with the lowest molecular weight (7.4 kg mol$^{-1}$). On the other hand, the polymer formed by **Ni5** showed the broadest dispersity ($M_w/M_n$ = 22.2), which clearly indicates multi-site behavior (Figure 8). Moreover, given the high melting temperature of the polymer formed using **Ni5**/EtAlCl₂ ($T_m$ = 120.3 °C) (run 20, Table 4), it would seem likely that each active site produces the same type of PE incorporating a significant amount of uniformly distributed long-chain

branches (LCBs). As a general point, all the PEs produced using the **Ni1–Ni5/**EtAlCl$_2$ systems showed properties which closely resemble industrial-grade LLDPEs reported elsewhere [9,48,49]. More importantly, almost all the complexes produced PEs with molecular weights in the range $10^4$–$10^5$ g mol$^{-1}$, which are in high demand for blending purposes to promote miscibility [50].

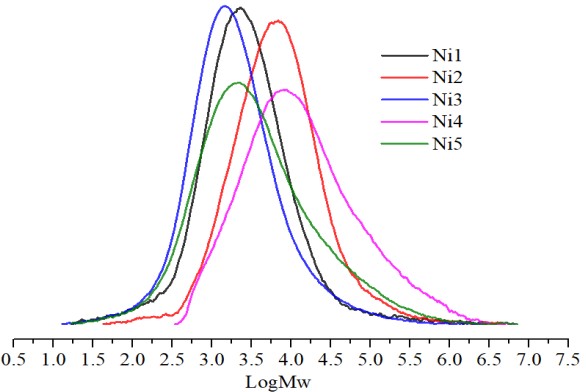

**Figure 8.** GPC curves of the polyethylene produced using **Ni1–Ni5/**EtAlCl$_2$ under optimum conditions (runs 3 and 17–20, Table 4).

### 2.2.3. Ethylene Polymerization Using **Ni1–Ni5** in the Presence of MMAO

With MMAO now employed as the aluminum co-catalyst, the performances of **Ni1–Ni5** were re-investigated as precatalysts for ethylene polymerization; the results are collected in Table 5. As with the EtAlCl$_2$ study, **Ni4** was chosen as the test precatalyst to identify an effective set of reaction parameters that would be later employed to screen the remaining precatalysts.

**Table 5.** Optimization of the polymerization conditions using **Ni4/**MMAO and catalytic evaluation of **Ni1–Ni3** and **Ni5** [a].

| Run | Precat. | T (°C) | t (min) | Al:Ni | Activity [b] | $M_w$ [c] | $M_w/M_n$ [c] | $R_i$ [d] | $R_t$ [e] | $T_m$ (°C) [f] |
|---|---|---|---|---|---|---|---|---|---|---|
| 1 | **Ni4** | 30 | 30 | 1250 | 1.6 | 8.2 | 4.1 | 114.1 | 1600.0 | 84.8, 114.4 |
| 2 | **Ni4** | 30 | 30 | 1500 | 2.0 | 10.1 | 4.5 | 143.3 | 1782.2 | 91.8, 113.3 |
| 3 | **Ni4** | 30 | 30 | 1750 | 2.4 | 8.5 | 5.1 | 171.8 | 2880.0 | 84.8, 114.5 |
| 4 | **Ni4** | 30 | 30 | 2000 | 2.1 | 7.2 | 3.4 | 152.6 | 1983.3 | 93.4, 110.6 |
| 5 | **Ni4** | 30 | 30 | 2250 | 1.6 | 6.0 | 2.9 | 116.2 | 1559.1 | 86.9, 114.1 |
| 6 | **Ni4** | 20 | 30 | 1750 | 1.2 | 21.6 | 8.9 | 88.4 | 1020.2 | 101.5, 114.0 |
| 7 | **Ni4** | 40 | 30 | 1750 | 2.2 | 7.6 | 5.5 | 153.3 | 3185.2 | 76.4, 115.4 |
| 8 | **Ni4** | 50 | 30 | 1750 | 0.9 | 3.9 | 4.2 | 61.3 | 1937.7 | 61.1, 119.8 |
| 9 | **Ni4** | 30 | 10 | 1750 | 1.6 | 6.2 | 3.3 | 111.1 | 1722.6 | 82.3, 113.9 |
| 10 | **Ni4** | 30 | 20 | 1750 | 1.7 | 6.5 | 3.4 | 119.9 | 1163.5 | 85.5, 113.1 |
| 11 | **Ni4** | 30 | 40 | 1750 | 1.9 | 9.6 | 4.3 | 136.2 | 1702.1 | 93.9, 112.3 |
| 12 | **Ni4** | 30 | 50 | 1750 | 1.8 | 10.3 | 5.5 | 140.9 | 1974.1 | 92.3, 111.8 |
| 13 | **Ni4** | 30 | 60 | 1750 | 1.6 | 11.4 | 5.3 | 233.9 | 1487.7 | 95.1, 113.5 |
| 14 [g] | **Ni4** | 30 | 30 | 1750 | 0.8 | 5.5 | 2.8 | 57.8 | 814.5 | 70.9, 121.4 |
| 15 [h] | **Ni4** | 30 | 30 | 1750 | trace | - | - | - | - | - |
| 16 | **Ni1** | 30 | 30 | 1750 | 0.7 | 7.0 | 5.0 | 50.6 | 1014.3 | 85.3, 117.0 |
| 17 | **Ni2** | 30 | 30 | 1750 | 2.7 | 2.6 | 2.4 | 178.2 | 4984.6 | 82.9, 112.1 |
| 18 | **Ni3** | 30 | 30 | 1750 | 1.5 | 10.0 | 3.7 | 108.4 | 1125.1 | 90.2, 105.7 |
| 19 | **Ni5** | 30 | 30 | 1750 | 1.0 | 1.5 | 7.2 | 73.4 | 9903.8 | 69.0, 125.3 |

[a] Conditions: 2.0 μmol nickel precatalyst, 100 mL toluene, and 10 atm of C$_2$H$_4$; [b] $10^6$ g of PE (mol of Ni)$^{-1}$ h$^{-1}$; [c] $M_w$: kg mol$^{-1}$, determined by GPC; [d] Rate of monomer insertion, in units of mmol/h; [e] Rate of chain termination, in units of μmol/h; [f] Determined by DSC; [g] 5 atm of C$_2$H$_4$; [h] 1 atm of C$_2$H$_4$.

To begin with, the influence of the Al:Ni molar ratio was probed by performing the polymerization runs using **Ni4/**MMAO at ratios between 1250:1 and 2250:1 (runs 1–5,

Table 5). With the temperature and time fixed at 30 °C and 30 min, respectively, the catalytic activity reached a maximum of $2.4 \times 10^6$ g PE (mol of Ni)$^{-1}$ h$^{-1}$ at 1750:1. Further increasing the amount of MMAO from 1750:1 to 2250:1 reduced the catalytic activity of **Ni4** reaching its lowest value of $1.6 \times 10^6$ g PE (mol of Ni)$^{-1}$ h$^{-1}$ at 2250:1 (run 5, Table 5). On the other hand, the molecular weight reached a peak value of 10.1 kg mol$^{-1}$ at 1500:1 and then reduced at higher amounts attaining a minimum of 6.0 kg mol$^{-1}$ at 2250:1. Furthermore, there was some evidence for variations in the polymer dispersity with a maximum of 5.1 attainable at 1750:1 which then narrowed as the ratios increased (Figure S2). It would seem likely that chain termination via chain transfer to aluminum becomes more significant at larger Al:Ni molar ratios leading to lower molecular weight and dispersity [30].

Subsequently, the activity/temperature profile of **Ni4**/MMAO was examined by screening this catalyst at temperatures from 20 °C to 50 °C (runs 3 and 6–8, Table 5). Between 20 °C and 30 °C, the catalytic activity markedly increased from $1.2 \times 10^6$ g PE (mol of Ni)$^{-1}$ h$^{-1}$ to $2.4 \times 10^6$ g PE (mol of Ni)$^{-1}$ h$^{-1}$ and then gradually declined attaining a low point of $0.9 \times 10^6$ g PE (mol of Ni)$^{-1}$ h$^{-1}$ at 50 °C. Conversely, the molecular weight of the PEs lowered from the outset (21.6 kg mol$^{-1}$ at 20 °C) reaching a low point of 3.9 kg mol$^{-1}$ at 50 °C, which can be attributed to lower monomer concentration and gradual deactivation of the active species as the temperature was raised. These findings align well with the decreasing values of $R_i$ from 171.8 to 61.3 mmol/h when the polymerization temperatures are beyond optimum. By contrast, the values of $R_t$ showed some irregularities, which could be related to the poorer stability of the **Ni**/MMAO active species (runs 3, 7 and 8, Table 5) [44]. Indeed, similar sensitivity of the catalytic activity and molecular weight to reaction temperature has been observed previously for 2-(arylimino)pyridine- nickel [35], and $N,N$-diaryl-$\alpha$-diimine-nickel catalysts [40,51–53]. Furthermore, it was evident from the GPC traces that multi-modal features were characteristic of the polymers obtained between 20 °C and 40 °C, with a distinct lower molecular weight shoulder visible, which tended to disappear at 50 °C (Figure S3). Moreover, the DSC thermograms showed two separate peaks for each polymer sample, which would imply the presence of different active sites producing polymers differing in their chain length or branching leading to crystal lamellar with different thickness. Similar properties for the PEs have been reported for the aryl-bridged 2-iminopyridine nickel catalysts [35,54,55].

With the temperature fixed at 30 °C and the Al:Ni ratio at 1750:1, the response of **Ni4**/MMAO towards run time was studied by conducting polymerizations over 10, 20, 30, 40, 50 and 60 min (runs 3 and 9–13, Table 5). After 30 min, the catalytic activity reached its highest value of $2.4 \times 10^6$ g PE (mol of Ni)$^{-1}$ h$^{-1}$ and then slowly declined to $1.6 \times 10^6$ g PE (mol of Ni)$^{-1}$ h$^{-1}$ after one hour, indicating **Ni4**/MMAO displayed a similar induction period to **Ni4**/EtAlCl$_2$. On the other hand, the molecular weight of the PEs increased smoothly with time from 6.2 to 11.4 kg mol$^{-1}$, highlighting the capacity of this catalyst to promote chain propagation over prolonged reaction times. As for the polymer dispersity, moderately broad distributions were seen over shorter run times ($M_w/M_n$ range: 3.3–5.5), which became slightly multi-modal over longer run times (40–60 min) (see Figure S4). On lowering the ethylene feed pressure from 10 to 5 atm, nearly one-third of the catalytic activity was lost (run 3 vs. run 14, Table 5), while at 1 atm this catalyst was essentially inactive (run 14 vs. run 15, Table 5). Conversely, the molecular weight of the resulting PEs showed relatively little sensitivity to monomer pressure [19].

With an effective set of reaction conditions now in place for **Ni4**/MMAO (P$_{C2H4}$ = 10 atm, Al:Ni ratio = 1750:1, temperature = 30 °C and time = 30 min), the remaining nickel precatalysts, **Ni1**–**Ni3** and **Ni5**, were similarly evaluated (runs 3 and 16–19, Table 5). Inspection of the results revealed that the activities were generally less ($0.7$–$2.7 \times 10^6$ g PE (mol of Ni)$^{-1}$ h$^{-1}$) than for the **Ni**/EtAlCl$_2$ systems (2.6–4.7 g PE (mol of Ni)$^{-1}$ h$^{-1}$). Collectively, the catalytic activities decreased in the order of **Ni2** [2,6-di(Et)] > **Ni4** [2,6-di-(CH(4-F-C$_6$H$_4$)$_2$] > **Ni3** [2,6-di(CH(C$_6$H$_5$)$_2$)] > **Ni5** [2,6-di(*i*-Pr)-3-NO$_2$] > **Ni1** [2,6-di-(*i*-Pr)]. As with the EtAlCl$_2$ investigation, the less sterically hindered **Ni2** showed the highest catalytic activity of $2.7 \times 10^6$ g PE (mol of Ni)$^{-1}$ h$^{-1}$ (run 17, Table 5). Additionally, replacing the *ortho*-ethyl group with the sterically

more hindered isopropyl group resulted in a significant change in catalytic performance with the catalytic level decreasing from $2.7 \times 10^6$ to $0.7 \times 10^6$ g PE (mol of Ni)$^{-1}$ h$^{-1}$; related findings have been noted for bimetallic bis(arylimino)pyridine nickel precatalytsts [55]. By comparison of **Ni4** and **Ni3**, it is evident (and also noted in the EtAlCl$_2$ investigation) that the presence of fluorinated benzhydryl groups is beneficial to catalytic activity and polymer properties.

With respect to the polymers formed using **Ni1**–**Ni5**/MMAO, a narrower range in molecular weights was observed ($M_w$ range: 1.5–10.0 kg mol$^{-1}$) when compared to that seen using **Ni**/EtAlCl$_2$ ($M_w$ range: 7.4–61.5 kg mol$^{-1}$). On comparison of the polymers produced using *ortho*-ethyl **Ni2** and *ortho*-isopropyl **Ni1**, the molecular weight increased from 2.6 kg mol$^{-1}$ to 7.0 kg mol$^{-1}$, while the dual melting temperatures ($T_m$) of the polymers increased from 82.9/112.1 °C (**Ni2**/MMAO) to 85.3/117.0 °C (**Ni1**/MMAO) (runs 16 and 17, Table 5). These results suggest that the *ortho*-isopropyl groups provide good steric protection to the active metal center with the result that the insertion rate of the incoming monomer increases and the degree of branching decreases. As would be expected given the even greater hindrance about the metal center, the two precatalysts bearing either *ortho*-benzhydryl groups or its fluorinated derivative, **Ni3** and **Ni4**, generated the highest molecular weight polymers of the series. In terms of dispersity, *meta*-nitro **Ni5** gave the broadest dispersity ($M_w/M_n$ = 7.2) and the highest melting temperature ($T_m$ = 125.3 °C) among the MMAO-activated systems (run 19, Table 5 and Figure S5), in line with increased linearity of the polymer chain.

In brief, the catalytic activities observed using **Ni**/MMAO were less than those achieved by **Ni**/EtAlCl$_2$, as were the molecular weights and melting temperatures of the polymers. Evidently, these findings highlight not only the effects of ligand variations but also the importance of the precatalyst/co-catalyst combination with respect to both catalytic performance and properties of the resulting polymers.

### 2.2.4. Microstructural Studies of the Polyethylene

As has already been noted, all polymers generated using **Ni1**–**Ni5** in combination with EtAlCl$_2$ exhibited high $T_m$ values falling in the range 117.8–126.0 °C (Table 4), which is supportive of high linearity together with uniformly distributed branching leading to crystals with significant lamellae thickness. Significantly, similar melting temperatures have been reported for industrial grade LLDPE [9,48,56]. By contrast, in the MMAO-activated systems, each polymer sample possessed two $T_m$ values, in which the lower $T_m$ ranged from 61.1 to 101.5 °C and the higher one ranged from 105.7 to 125.3 °C (Table 5). These results suggest that higher branching and poorly ordered shorter chains contribute to the formation of crystals with different lamellar thickness. Indeed, structurally related 2-(arylimino)pyridine nickel precatalysts have recently been reported to display similar dual melt temperatures for their PEs [35].

Given the differences noted above, it was of interest to investigate the microstructural properties of these distinct types of PEs. Hence, two representative polymer samples prepared using **Ni4**/EtAlCl$_2$ at 30 °C (run 3, Table 4) and **Ni4**/MMAO at 30 °C (run 3, Table 5) were examined by $^1$H NMR and $^{13}$C NMR spectroscopy. To allow suitable solubility, the samples were dissolved in deuterated 1,1,2,2-tetrachloroethane-d$_2$ (C$_2$D$_2$Cl$_4$) at 100 °C and their spectra recorded at a comparable temperature. Firstly, the $^1$H NMR spectrum of the PE sample obtained using **Ni4**/EtAlCl$_2$ revealed the presence of both vinyl (–CH=CH$_2$) and vinylene (–CH=CH–) groups as the main types of unsaturation (Figure 9). In particular, the downfield signals at δ 5.89 (H$_b$) and δ 5.01–5.11 (H$_a$) with an integral ratio of 1:2 were characteristic of a vinyl end-group. Conversely, the signal for the vinylene group can be observed as a broad resonance at δ 5.48 (H$_c$/H$_{c'}$). Moreover, the vinylene to vinyl ratio was 5.3:1, which highlights not only the importance of β-hydride elimination but also double bond isomerization in these polymerizations; agostic interactions involving γ-, δ- and higher-hydrogens have also been invoked to account for the formation of vinylene functionalities [55–57].

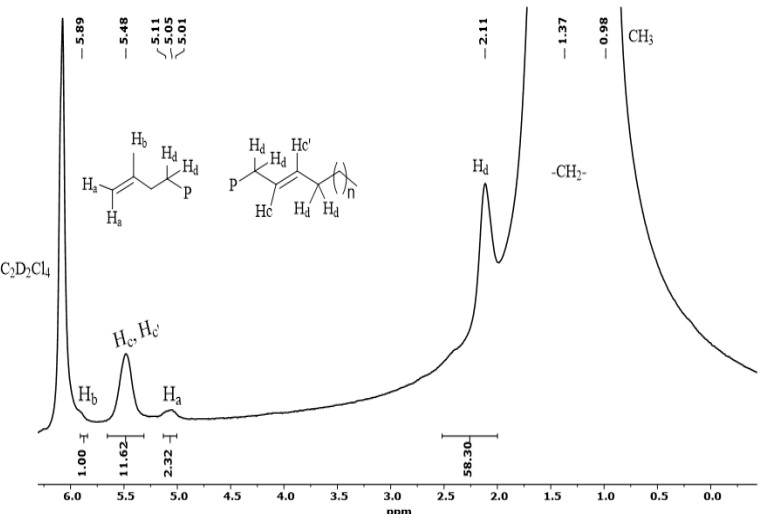

**Figure 9.** $^1$H NMR spectrum of the polyethylene sample generated using **Ni4**/EtAlCl$_2$ at 30 °C (run 3, Table 4); recorded at 100 °C in $d$-C$_2$D$_2$Cl$_4$.

The more upfield peaks in the $^1$H NMR spectrum at δ 1.37, 2.11 and 0.98 can be assigned to the main chain protons (–CH$_2$–)$_n$, allylic protons (H$_d$) and methyl protons, respectively [34]. Further support for the assignment of unsaturation was provided by the $^{13}$C NMR spectrum, which showed the characteristic downfield signals (a, b, c and c′) for the corresponding vinyl and vinylene unsaturated carbons (Figure 10).

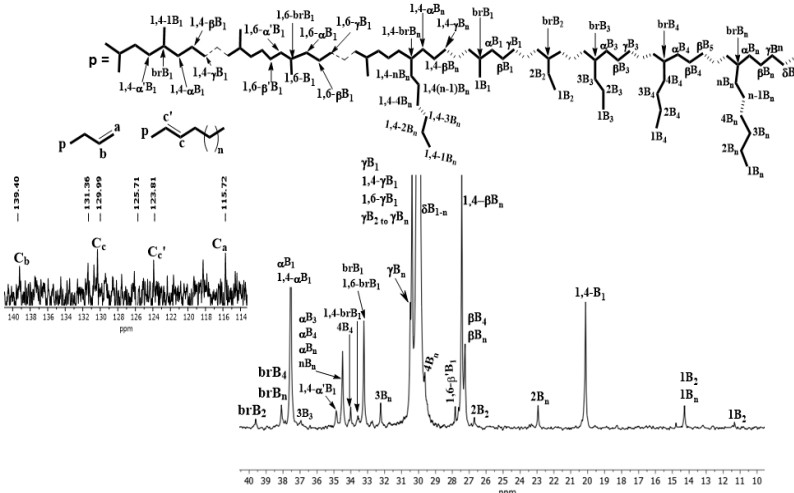

**Figure 10.** $^{13}$C NMR spectrum of the polyethylene sample produced using **Ni4**/EtAlCl$_2$ at 30 °C (run 3, Table 4), including an inset of the alkenic region and a segment of the assigned polymer backbone; recorded at 100 °C in $d$-C$_2$D$_2$Cl$_4$.

Furthermore, an estimation of the branching density and branching distribution in this polymer sample (run 3, Table 4) was determined by analysis of the $^{13}$C NMR spectrum using approaches described in the literature [24,25,58,59]. In short, this polymer contained a moderate branching density of 48/1000 Cs, including methyl (78.41%), ethyl (2.91%), propyl (5.64%), butyl (4.41%) and LCBs (8.63%) (Figure 9 and Tables S2 and S3). There was no evidence of isomerism on the side chains to form hyperbranched chains, which is consistent with previous observations [60].

Similarly, the $^1$H NMR spectrum of the PE obtained by **Ni4**/MMAO was also investigated (run 3, Table 5). As seen earlier, this spectrum showed the existence of both vinyl and vinylene unsaturated groups with the (–CH=CH–) to (–CH=CH$_2$) ratio now at 9.9:1, which

is nearly twice as high as that seen with the EtAlCl$_2$-activated system (Figure S7). Other upfield signals appeared between 2.14–0.94 ppm can be assigned to the allylic protons (CH$_2$=CH–CHR–) and the main backbone protons in line with previous findings [23]. In terms of the branching density and type of branch, the $^{13}$C NMR spectrum of this polymer sample (run 3, Table 5) was again informative exhibiting a relatively higher branching density of 66.3 branches per 1000 Cs including methyl (82.82%), ethyl (0.98%), propyl (6.42%), butyl (2.98%) and LCBs (6.80%) (Figure S8, Tables S4 and S5).

As a key point, when the activator was switched from EtAlCl$_2$ to MMAO, the LCBs and butyl branches in both polymers decreased, respectively, from 8.63% to 6.80% and 4.41% to 2.98%. On the other hand, the number of methyl branches increased from 78.41% to 82.82%. Based on literature precedent, we propose the methyl branch is generated by a single chain walking step involving β-hydride elimination, 2,1-reinsertion and then coordination/insertion of ethylene [34,37,61]. Further chain walking accounts for the longer chain branching, which is evidently favored by the **Ni**/EtAlCl$_2$ system. Similar conclusions were recently reached based on experimental results [37,61–64], and computational simulations [65]. As an alternative pathway, Gao and co-workers have proposed that LCBs can be formed by ethylene insertion into a primary Ni-alkyl species, which originates from the migration of nickel to a methyl terminal of the growing chain [62].

### 2.2.5. Comparison between Current and Reported Analogues

When compared with previous work, two noteworthy findings can be identified with the current systems. Firstly, all precatalysts (**Ni1**–**Ni5**) catalyzed ethylene to form PEs as the only type of product without contamination with oligomeric products, a finding that is unlike previous reports using 2-(arylimino)pyridine-nickel catalysts that have a tendency to afford oligomers or mixed products [56,66]. Secondly, the sterically crowded *ortho*-isopropyl precatalyst **Ni1** exhibited the lowest catalytic activity (Tables 3 and 4), which is quite different to that observed for similarly substituted *N*-aryl substituted nickel catalysts [23,67,68].

To allow a broader comparison, the catalytic behavior and polymer properties exhibited by representative precatalyst **Ni4** (**E**, Figure 1) were compared with structurally related **A**–**D** (Figure 11). In particular, the data collected for **A**–**E** were based on the use of the most effective co-catalyst at the highest effective operating temperature. Inspection of Figure 11 reveals **Ni4**/EtAlCl$_2$ at 50 °C (**E**-50$_{EtAlCl2}$, run 8, Table 4) showed not only higher activity but also produced polymers with the highest molecular weight. Specifically, the relative catalytic activities were found to fall as follows: **E**-50$_{EtAlCl2}$ > **E**-60$_{EtAlCl2}$ > **C**-50$_{MAO}$ > **D**-50$_{MAO}$ > **B**-50$_{MAO}$. Furthermore, the molecular weight ($M_w$ = 32.9 kg mol$^{-1}$) and dispersity ($M_w/M_n$ = 18.2) of the polymer produced using **E** were both nearly ten times higher than the corresponding values seen for **A**–**D**. Regarding the $T_m$ values of the PEs, that produced using **E** again ranked at the top end of the range: **E**-50$_{EtAlCl2}$ (123.7 °C) > **C**-50$_{MAO}$ (62.6 °C) > **B**-50$_{MAO}$ (10.1 °C). As a further point, **E**-30$_{EtAlCl2}$, evaluated under optimized conditions (run 3, Table 4), exhibited catalytic activity (3.8 × 10$^6$ g PE (mol of Ni)$^{-1}$ h$^{-1}$) that exceeds levels seen for a number of structurally nickel systems under comparable conditions [22,32,35,36]. Likewise, the molecular weight ($M_w$ = 61.5 kg mol$^{-1}$) and melting temperature ($T_m$ = 125.9 °C) of the corresponding polymer were much higher. As a final more general point, the polymers generated in this work with either **Ni**/EtAlCl$_2$ of **Ni**/MMAO suggested multi-site characteristics and produced vinyl and vinylene unsaturated PEs with an obvious preference for internal unsaturation. By stark contrast, unimodal and vinyl terminated PEs have been a characteristic of the polymers obtained using previously reported 2-(arylimino)pyridine nickel catalysts [34–36].

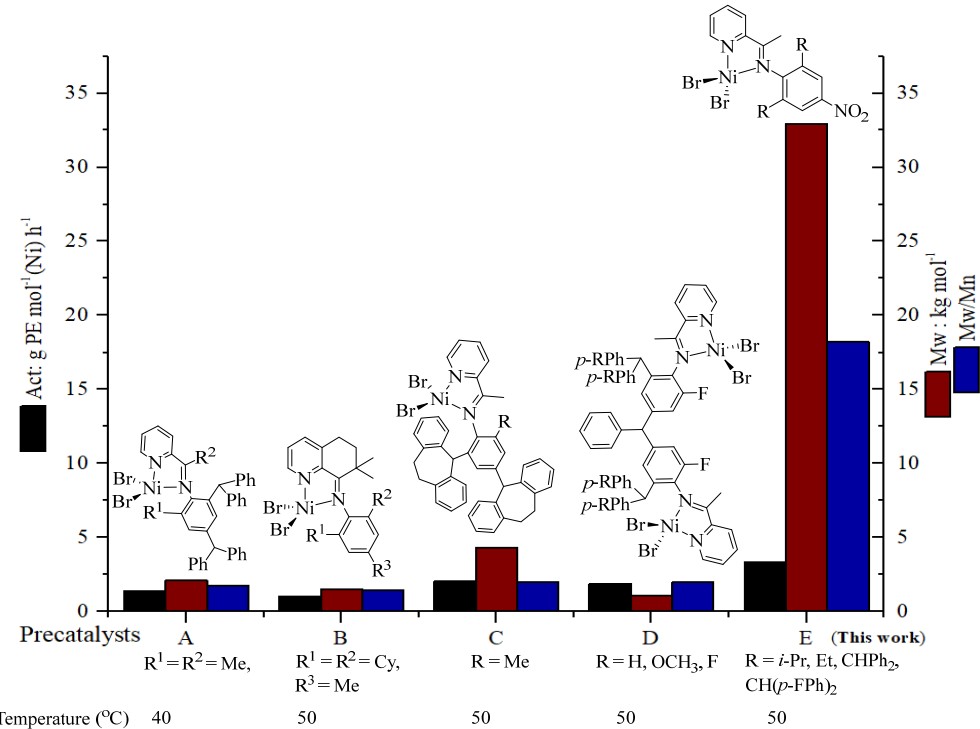

**Figure 11.** Comparison of the catalytic performance of **Ni4** (**E**, Scheme 1) with previously reported, **A–D** (Scheme 1), at elevated polymerization temperatures (T = 40 °C for **A** and 50 °C for **B–E**), $P_{C2H4}$ = 10 atm in presence of their optimum co-catalyst.

## 3. Experimental Section

### 3.1. General Consideration

All manipulations involving air and/or moisture-sensitive compounds were undertaken under an inert nitrogen atmosphere using standard Schlenk techniques. Toluene was heated to reflux over sodium and distilled under nitrogen before use. Methylaluminoxane (MAO, 1.46 M in toluene) and modified methylaluminoxane (MMAO, 1.93 M in toluene, containing 20–25% Al(*i*-Bu)₃) were purchased from Anhui Botai Electronic Materials Co., Ltd., whereas ethylaluminum dichloride (EtAlCl₂, 2.17 M in hexane), ethylaluminum sesquichloride (EASC, 0.87 M in toluene), and diethylaluminum chloride (Et₂AlCl, 1.17 M in heptane) were purchased from Lianli Chemical (Yantai, China). High purity ethylene was purchased from Beijing Yanshan Petrochemical Co. and used as received. Other reagents were purchased from Aldrich, Acros, or local suppliers (Beijing, China). NMR spectra of the ligands were recorded on a Bruker DMX 400 MHz instrument (Bruker, Fällanden, Switzerland) at ambient temperature using TMS as the internal standard, while the PEs were recorded on a Bruker DMX 500 MHz instrument (Bruker, Fällanden, Switzerland) at 100 °C in deuterated 1,1,2,2-tetrachloroethane–*d₂* (C₂D₂Cl₄) with TMS as an internal standard. FTIR spectra were conducted on a Perkin–Elmer System 2000 FT-IR spectrometer (Perkin-Elmer, Waltham, MA, USA). Elemental analysis was carried out using a Flash EA 1112 micro-analyzer (Thermo Fisher Scientific, Waltham, MA, USA). The molecular weight and polymer dispersity of the PEs were determined using a PL-GPC220 instrument (Beijing, China) at 150 °C with 1,2,4-trichlorobenzene as solvent. The melting points ($T_m$) of the PEs were measured from the second scanning run on a Perkin–Elmer TA-Q2000 (Perkin-Elmer, Waltham, MA, USA) differential scanning calorimetry (DSC) analyzer under a nitrogen atmosphere. In the procedure, a sample of 4.0–6.0 mg was heated from −20 °C to 160 °C at a rate of 10 °C min⁻¹.

*3.2. Experimental Procedure for the Synthesis of **L1–L5***

3.2.1. 2-{(2,6-*i*-Pr$_2$-4-NO$_2$C$_6$H$_2$)N=CMe}C$_5$H$_4$N (**L1**)

A round-bottomed flask equipped with a stir bar and a Dean–Stark trap was loaded with a mixture of 2-acetylpyridine (0.25 g, 2.06 mmol) and 2,6-diisopropyl-4-nitroaniline (0.50 g, 2.20 mmol) in toluene (50 mL). The mixture was stirred and heated to reflux for half an hour until the formation of a homogeneous solution. *p*-Toluenesulfonic acid (15 mol%) was added as catalyst and the reaction mixture further heated at reflux for 10 h. Upon cooling to room temperature, all volatiles were removed under reduced pressure and the residue purified by column chromatography on alumina using petroleum ether/ethyl acetate (250:1) as the eluent. **L1** was isolated as a yellow powder (0.53 g, 79%). FT-IR (cm$^{-1}$): 2962 (m), 2929 (w), 2870 (w), 1652 (*v*(C=N), m), 1583 (m), 1567 (w), 1511 (s), 1464 (m), 1434 (m), 1363 (s), 1342 (s), 1323 (s), 1302 (s), 1257 (s), 1240 (s), 1201 (m), 1148 (w), 1101 (s), 1045 (w), 993 (w), 965 (w), 901 (w), 808 (w), 785 (m), 756 (m), 746 (m), 720 (m). $^1$H NMR (400 Hz, CDCl$_3$, TMS): δ 8.79–8.7.77 (d, J = 8.0 Hz, 1H, Py–H), 8.33–8.30 (d, J = 12.0 Hz, 1H, Py–H), 8.06 (s, 2H, Ar–H), 7.87–7.82 (t, J = 10.0 Hz, 1H, Py–H), 7.46–7.42 (t, J = 8.0 Hz, 1H, Ar–H), 2.84–2.71 (s, 2H, –CH–), 2.23 (s, 3H, –CH$_3$), 1.22–1.16 (m, 12H, –CH$_3$). $^{13}$C NMR (100 MHz, CDCl$_3$, TMS): δ 162.8, 160.4, 150.7, 141.4, 138.2, 138.2, 130.6, 130.5, 129.4, 127.8, 123, 118.4, 115.3, 115.1, 49.3, 26.9, 24.4, 20.8, 12.8, 12.6. Anal. Calcd. for C$_{19}$H$_{23}$N$_3$O$_2$ (325.41): C, 70.13, 7.12; N, 12.91. Found: C, 69.78; H, 7.13; N, 12.79.

3.2.2. 2-{(2,6-Et$_2$-4-NO$_2$C$_6$H$_2$)N=CMe}C$_5$H$_4$N (**L2**)

In a manner similar that described for the synthesis of **L1**, **L2** was isolated as a yellow powder (0.50 g, 83%). FT-IR (cm$^{-1}$): 3052 (w), 2964 (w), 2933 (w), 2872 (w), 1652 (*v*(C=N), m), 1584 (m), 1505 (s), 1460 (m), 1434 (m), 1365 (m), 1334 (s), 1301 (m), 1259 (s), 1242 (m), 1204 (m), 1096 (s), 1040 (m), 879 (m), 799 (s), 780 (s), 741 (s), 710 (s). $^1$H NMR (400 MHz, CDCl$_3$, TMS): δ 8.69–8.68 (d, J = 4.0 Hz, 1H, Py–H), 8.32–8.30 (d, J = 8.0 Hz, 1H, Py–H), 8.02 (s, 2H, Ar–H), 7.85–7.81 (t, J = 8.0 Hz, 1H, Py–H), 7.44–7.40 (t, J = 8.0 Hz, 1H, Py–H), 2.49–2.35 (m, 4H, –CH$_2$–), 2.21 (s, 3H, CH$_3$), 1.21–1.17 (m, 6H, CH$_3$). $^{13}$C NMR (100 MHz, CDCl$_3$, TMS): δ 166.4, 154.4, 152.9, 147.8, 143.0, 135.6, 131.4, 124.4, 120.6, 120.4, 23.5, 16.5, 12.0. Anal. Calcd. for C$_{17}$H$_{19}$N$_3$O$_2$ (297.36): C, 68.67.; H, 6.44; N, 14.13. Found: C, 69.03; H, 6.44; N, 13.88.

3.2.3. 2-{(2,6-((C$_6$H$_5$)$_2$CH)$_2$-4-NO$_2$C$_6$H$_2$)N=CMe}C$_5$H$_4$N (**L3**)

In a manner similar that described for the synthesis of **L1**, **L3** was isolated as an orange powder (0.77 g, 65%). FT-IR (cm$^{-1}$): 3058 (w), 3025 (w), 1966 (w), 1658 (*v*(C=N), m), 1581(w), 1513 (s), 1493 (m), 1470 (w), 1448 (w), 1434 (w), 1363 (w), 1329 (s) 1297 (m), 1262 (w), 1232 (m), 1151 (w), 1093 (m), 1027 (w), 996 (w), 970 (w), 917 (w), 870 (w), 787 (m), 767 (m), 740 (m), 698 (s). $^1$H NMR (400 MHz, CDCl$_3$, TMS): δ 8.62–8.60 (d, J = 8.0 Hz, 1H, Py–H), 7.98–7.96 (d, J = 8.0 Hz, 1H, Py–H) 7.78–7.73 (m, 2H, Py–H), 7.40–7.37 (t, J = 6.0 Hz, 1H, Ar–H), 7.28–7.19 (m, 8H, Ar–H), 7.02–6.97 (m, 8H, Ar–H), 5.30 (s, 2H, –CH–), 1.00 (s, 3H, –CH$_3$). $^{13}$C NMR (100 MHz, CDCl$_3$, TMS): δ 169.7, 154.7, 154.3, 142.0, 140.9, 136.4, 134.0, 129.7, 129.3, 128.8, 128.5, 126.9, 126.8, 125.3, 123.6, 121.6, 52.2, 17.6. Anal. Calcd. for C$_{39}$H$_{31}$N$_3$O$_2$ (573.70): C, 81.50; H, 5.45; N, 7.32. Found: C, 81.34; H, 5.26; N, 7.18.

3.2.4. 2-{(2,6-((4-F-C$_6$H$_4$)$_2$CH)$_2$-4-NO$_2$C$_6$H$_2$)N=CMe}C$_5$H$_4$N (**L4**)

In a manner similar that described for the synthesis of **L1**, **L4** was isolated as a yellow powder (0.79 g, 60%). FT-IR (cm$^{-1}$): 3044 (w), 3003 (w), 2925 (w), 1643 (*v*(C=N), m), 1602 (m), 1582 (m), 1504 (s), 1465 (w), 1431 (w), 1364 (s), 1334 (w), 1300 (m), 1217 (s), 1158 (s), 1096 (m), 1016 (w), 962 (w), 913 (w), 880 (w), 835 (s), 791 (m), 744 (m), 717 (m), 688 (m). $^1$H NMR (400 MHz, CDCl$_3$, TMS): δ 8.64–8.63 (s, J = 4.0 Hz, 1H, Py–H), 7.94–7.92 (d, J = 8.0 Hz, 1H, Py–H), 7.79–7.75 (t, J = 8.0 Hz, 3H, Py–H), 7.43–7.40 (t, J = 6.0 Hz, 1H, Ar–H), 6.98–6.92 (m, 16H, Ar–H), 5.25 (s, 2H, –CH–), 1.18 (s, 3H, –CH$_3$). $^{13}$C NMR (100 MHz, CDCl$_3$, TMS): δ 169.6, 163.0, 162.9, 160.5, 160.5, 154.4, 153.9, 149.0, 143.7, 137.4, 137.4, 136.5, 136.5, 136.5, 133.8, 131.0, 130.9, 130.6, 130.5, 125.6, 123.5, 121.4,115.9, 115.7, 115.6, 115.3, 50.7,

17.9. Anal. Calcd. for $C_{39}H_{27}F_4N_3O_2$ (645.66): C, 72.55; H, 4.22; N, 6.51. Found: C, 71.85; H, 4.24; N, 6.33.

### 3.2.5. 2-{(2,6-*i*-Pr$_2$-3-NO$_2$-4-(4-F-C$_6$H$_4$)C$_6$H)=CMe}C$_5$H$_4$N (**L5**)

In a manner similar that described for the synthesis of **L1**, but with petroleum ether and ethyl acetate (50/1 v/v) used as the eluent for column chromatography, **L5** was isolated as an orange powder (0.42 g, 39%). FT-IR (cm$^{-1}$): 3067 (w), 3089 (w), 2963 (w), 2925 (w), 2869 (w), 1650 ($v$(C=N), m), 1598 (m), 1567 (w), 1524 (s), 1503 (s), 1465 (m), 1444 (w), 1363 (s), 1302 (w), 1223 (s), 1197 (m), 1157 (m), 1101 (m), 1043 (w), 1014 (w), 967 (w), 896 (w), 823 (w), 784 (w), 741 (w), 673 (w). $^1$H NMR (400 MHz, CDCl$_3$, TMS): δ 8.69–8.68 (d, *J* = 4.0 Hz, 1H, Py–H), 8.33–8.31 (d, *J* = 8.0 Hz, 1H, Py–H), 7.85–7.81 (t, *J* = 8.0 Hz, Ar–1H), 7.43–7.40 (t, *J* = 6.0 Hz, 1H), 7.07–6.98 (m, 8H, Ar–H), 6.83 (s, 1H, Ar–H$_m$), 5.43 (s, 1H, –CH–), 2.88–2.85 (m, 1H, –CH–), 2.64–2.60 (m, 1H, –CH–), 2.25 (s, 3H, –CH$_3$), 1.28–26 (d, *J* = 8.0 Hz, 3H, CH$_3$), 1.09–1.07 (d, *J* = 8.0 Hz, 3H, CH$_3$), 1.02–1.01 (d, *J* = 4 Hz, 3H, –CH$_3$), 0.96–0.94 (d, *J* = 8.0 Hz, 3H, CH$_3$). $^{13}$C NMR (100 MHz, CDCl$_3$, TMS): δ 168.6, 162.9, 160.5, 155.6, 150.1, 148.8, 147.1, 138.0, 138.0, 136.6, 130.6, 130.5, 128.6, 126.2, 125.8, 125.2, 121.4, 115.4, 115.4, 115.2, 115.2, 49.7, 29.5, 27.8, 23.3, 21.9, 21.3, 19.1, 18.2. Anal. Calcd. for $C_{32}H_{31}Br_2F_2N_3O_2$ (527.62): C, 72.85; H, 5.92; N, 7.96. Found: C, 72.50; H, 5.89; N, 7.87.

### 3.3. Experimental Procedure for the Synthesis of **Ni1–Ni5**

### 3.3.1. [2-{(2,6-i-Pr$_2$-4-NO$_2$C$_6$H$_2$)N=CMe}C$_5$H$_4$N]NiBr$_2$ (**Ni1**)

(DME)NiBr$_2$ (0.30 g, 0.98 mmol) and **L1** (0.36 g, 1.10 mmol) were combined in a Schlenk flask containing a stir bar and under a nitrogen atmosphere. An equal volume of dichloromethane and ethanol (20 mL, 1:1 v/v) was added and the reaction stirred at room temperature for 16 h. The supernatant liquid was decanted, and the residue repeatedly recrystallized from CH$_2$Cl$_2$ and Et$_2$O, washed with Et$_2$O (4 × 10 mL) and then dried to afford the product as a pale green powder (0.48 g, 91%). FT-IR (cm$^{-1}$): 2967 (w), 2927 m), 2872 (w), 1621 ($v$(C=N), w), 1597 (w), 1574 (w), 1514 (s), 1441 (m), 1376 (m), 1350 (m), 1322 (s), 1262 (m), 1192 (w), 1168 (w), 1112 (m), 1075 (m), 1050 (w), 1022 (w), 937 (w), 912 (w), 839 (w), 786 (s), 745 (m). Anal. Calcd. for $C_{19}H_{23}Br_2N_3O_2$ (543.91): C, 41.96; H, 4.26; N, 7.73. Found: C, 42.12; H, 4.64, N, 7.48.

### 3.3.2. [2-{(2,6-Et$_2$-4-NO$_2$C$_6$H$_2$)N=CMe}C$_5$H$_4$N]NiBr$_2$ (**Ni2**)

Using the same procedure and molar ratios as that described for **Ni1**, **Ni2** was isolated as a brown powder (0.47 g, 91%). FT-IR (cm$^{-1}$): 3163 (w), 2976 (w), 1626 ($v$(C=N), m), 1592 (m), 1516 (s), 1446 (m), 1320 (s), 1257 (s), 1193 (w), 1104 (w), 1054 (w), 1024 (w), 897 (m), 841 (w), 778 (s), 744 (s). Anal. Calcd. for $C_{34}H_{38}Br_4N_6Ni_2O_4{\cdot}2H_2O$ (533.87): C, 38.25; H, 3.96; N, 7.87. Found: C, 37.86; H, 4.25; N, 7.58.

### 3.3.3. [2-{(2,6-((C$_6$H$_5$)$_2$CH)$_2$-4-NO$_2$C$_6$H$_2$)N=CMe}C$_5$H$_4$N]NiBr$_2$ (**Ni3**)

Using the same procedure and molar ratios as that described for **Ni1**, **Ni3** was isolated as a green powder (0.88 g, 56%). FT-IR (cm$^{-1}$): 3279 (w), 3028 (w), 1975 (w), 1621 ($v$(C=N), m), 1597 (m), 1522 (s), 1494 (m), 1444 (m), 1398 (w), 1375 (w), 1334 (s), 1323 (m), 1257 (m), 1187 (w), 1075 (w), 1033 (m), 916 (w), 874 (w), 819 (w), 797 (w), 767 (m), 743 (m), 701 (s) 671 (m), 654 (m). Anal. Calcd. for $C_{120}H_{115}N_9Br_6Cl_6O_{14}Ni_3$ (2775.49): C, 51.93; H, 4.18; N, 4.54. Found: C, 51.74; H, 4.11; N, 4.36.

### 3.3.4. [2-{(2,6-((4-F-C$_6$H$_4$)$_2$CH)$_2$-4-NO$_2$C$_6$H$_2$)N=CMe}C$_5$H$_4$N]NiBr$_2$ (**Ni4**)

Using the same procedure and molar ratios as that described for **Ni1**, **Ni4** was isolated as a brown powder (0.68 g, 80%). FT-IR (cm$^{-1}$): 3317 (w), 3079 (w), 2911 (w), 1598 ($v$(C=N), m), 1505 (s), 1438 (w), 1343 (m), 1317 (m), 1224 (s), 1158 (s), 1098 (w), 1019 (w), 914 (w), 881 (w), 839 (s), 980 (m), 740 (w), 669 (w). Anal. Calcd. for $C_{39}H_{27}Br_2F_4N_3NiO_2$ (864.16): C, 54.21; H, 3.5; N, 4.86. Found: C, 54.04; H, 3.25; N, 4.57.

### 3.3.5. [2-{(2,6-i-Pr$_2$-3-NO$_2$-4-(4-F-C$_6$H$_4$)$_2$C$_6$H)N=CMe}C$_5$H$_4$N]NiBr$_2$ (**Ni5**)

Using the same procedure and molar ratios as that described for **Ni1**, **Ni5** was isolated as a brown powder (0.63 g, 94%). FT-IR (cm$^{-1}$): 2978 (w), 2935 (w), 1618 ($v$(C=N), m), 1593 (w), 1536 (w), 1504 (s), 1464 (w), 1445 (w), 1372 (m), 1328 (w), 1319 (w) 1261 (w), 1218 (s), 1189 (w), 1159 (m), 1128 (w), 1094 (w), 1049 (w), 1024 (w), 957 (w), 917 (w), 865 (w), 846 (m), 823 (m), 775 (m), 743 (w), 709 (w), 653 (w). Anal. Calcd. for C$_{32}$H$_{31}$Br$_2$F$_2$N$_3$NiO$_2$ (746.13): C, 51.65; H, 4.20; N, 7.53. Found: C, 51.72, H, 4.31; N, 7.39.

### 3.4. Ethylene Polymerization Evaluation

#### 3.4.1. At P$_{C2H4}$ = 1 atm

2 μmol of **Ni4** was loaded into a 250 mL Schlenk tube, equipped with a stir bar, and the tube evacuated and backfilled with nitrogen twice and ethylene once. Under an ethylene atmosphere, toluene (30 mL) followed by a pre-determined amount of aluminum alkyl co-catalyst (EtAlCl$_2$, MMAO) added by syringe. The reaction mixture was then stirred at the required temperature at P$_{C2H4}$ = 1 atm for 30 min before the ethylene supply was disconnected and the vessel vented. The resulting mixture was then quenched with 15% hydrochloric acid in ethanol and no polymer product was detected.

#### 3.4.2. At P$_{C2H4}$ = 5 or 10 atm

The ethylene polymerization process was carried out in a stainless-steel autoclave (250 mL) fitted with a pressure control system, temperature controller and mechanical stirrer. The autoclave was evacuated and backfilled with nitrogen three times and then with ethylene once. When the reactor had reached the desired reaction temperature, toluene (25 mL) and a solution of the nickel precatalyst dissolved in toluene (25 mL) were added successively. The required amount of aluminum co-catalyst and remaining toluene (50 mL) was then added and the mixture stirred at either P$_{C2H4}$ = 10 or 5 atm for the selected time. At the end of the run, the supply of ethylene was ceased, and the reactor vented. The resulting mixture was quenched with 15% solution hydrochloric acid in ethanol and filtered. After drying for 8 h at 50 °C under reduced pressure, the polymer was weighed.

### 3.5. X-ray Crystallographic Studies

Single crystals of **Ni3**, **Ni4** and **Ni5** suitable for the X-ray determinations were grown as described in the results and discussion section. The X-ray studies were performed on an XtaLAB Synergy-R HyPix diffractometer with mirror monochromatic Cu-Kα radiation (λ = 1.54184 Å) at 169.99 K; cell parameters were obtained by global refinement of the position of all collected reflections. The intensities were corrected for Lorentz and polarization effects and empirical absorption. All hydrogen atoms were placed in calculated positions. Structure determination by direct methods (and Patterson methods) was performed on Olex2 [69] by using the SHELXT (Sheldrick, 2015) [70], and structure refinement by using full-matrix least-squares on $F^2$ was performed by using SHELXL (Sheldrick, 2015) [71]. The free solvent molecules were squeezed with PLATON software [72,73]. Representations of the structures were generated with the ORTEP program [74]. The details of the X-ray structure determination data and refinements for **Ni3**, **Ni4** and **Ni5** are provided in the supporting information (Table S1).

## 4. Conclusions

In summary, a series of *para*-nitro containing 2-{(2,6-R$_2$-4-nitro-phenylimino) ethyl}pyridine-nickel(II) bromide (R = *i*-Pr **Ni1**, Et **Ni2**, CH(C$_6$H$_5$)$_2$ **Ni3**, CH(4-F-C$_6$H$_4$)$_2$ **Ni4**) complexes and a *meta*-nitro comparator (**Ni5**) have been successfully synthesized and characterized. The molecular structures of Ni3·xH$_2$O (x = 2, 3), **Ni4** and **Ni5** were studied by single-crystal X-ray diffraction. Upon activation with EtAlCl$_2$, all complexes displayed very good catalytic activities for ethylene polymerization with levels reaching up to 4.7 × 10$^6$ g PE (mol of Ni)$^{-1}$ h$^{-1}$ for **Ni2**/EtAlCl$_2$ at 30 °C with minimal depreciation in performance at 60 °C. The resulting polymers exhibited an assortment of attractive properties, including a broad range in molecular weights

(range: 7.4–137.7 kg mol$^{-1}$), dispersities ($M_w/M_n$ range: 2.2–24.4), and a significant amount of uniformly distributed long chain branches. Indeed, these polymeric materials resemble industrial-grade LLDPEs, which are in high demand in industry for various applications, including for blending processes due to their high miscibility. On the other hand, the PE's formed using **Ni1–Ni4**/MMAO were of lower molecular weight and displayed higher levels of branching. Notably, *meta*-nitro **Ni5** was the only precatalyst that generated PEs with high linearity on activation with either EtAlCl$_2$ or MMAO. In terms of the chain growth, both coordination/insertion and chain walking are competitive with the type of aluminum activator influential on the degree of chain walking and in turn the amounts of long chain branching. In terms of chain-termination, β-hydride elimination and chain isomerization are operational resulting in (–CH=CH–)/(–CH=CH$_2$) ratios of 5.3:1 with EtAlCl$_2$ and 9.9:1 using MMAO. Overall, these results highlight how the aluminum co-catalyst can play a key role on not only the catalytic activity of nickel catalysts but also on the properties of the polymer.

**Supplementary Materials:** The following are available online at https://www.mdpi.com/article/10.3390/catal12090961/s1, Scheme S1; Synthetic route of anilines (A3 and A4); S2: Synthetic route of 2,6-diisopropyl-3-nitro-4-(4,4-difluorobenzhydryl)aniline (A5); Figure S1: The three independent molecules (A, B and C) present in the unit cell of Ni3·xH$_2$O (x = 2, 3) including three molecules of dichloromethane; Figure S2: GPC curves of the polyethylene produced using Ni4/MMAO at different molar ratios; Figure S3: GPC curves of the polyethylene produced using Ni4/MMAO at different run temperatures; Figure S4: GPC curves of the polyethylene produced using Ni4/MMAO over different run times; Figure S5: GPC curves of the polyethylene produced using Ni1–Ni5/MMAO under optimum conditions; Figure S6: DSC thermogram of the polyethylene produced using Ni4/MMAO (run 4, Table 5); Figure S7: $^1$H NMR spectrum of the polyethylene sample produced using Ni4/MMAO recorded at 100 °C in *d*-C$_2$D$_2$Cl$_4$; Figure S8: $^{13}$C NMR spectrum of the polyethylene sample produced using Ni4/MMAO recorded at 100 °C in *d*-C$_2$D$_2$Cl$_4$; Figures S9–S21: $^1$H/$^{13}$C NMR and FTIR spectrum of anilines, ligands and complexes; Table S1: Crystallographic data for Ni3·xH$_2$O (x = 2, 3), Ni4 and Ni5; Table S2: $^{13}$C NMR data of the polyethylene sample produced using Ni4/EtAlCl$_2$ at 30 °C (run 3, Table 4); Table S3: Branching analysis for the polyethylene sample produced using Ni4/EtAlCl$_2$. Table S4: $^{13}$C NMR data for the polyethylene sample produced using Ni4/MMAO. Table S5: Branching analysis for the polyethylene sample produced using Ni4/MMAO [39,75–78]. Crystallographic information for **Ni3** (CCDC 2195148), **Ni4** (CCDC 2195149) and **Ni5** (CCDC 2195150).

**Author Contributions:** Design of the study by W.-H.S.; synthesis of the organic compounds and the nickel complexes by D.D.S.; characterization by D.D.S., Q.Z., M.L. and W.-H.S.; X-ray study by Q.Z. and Y.S.; catalytic study by D.D.S.; characterization of polyethylene by D.D.S.; writing and editing D.D.S., Q.Z., G.A.S. and W.-H.S. All authors have read and agreed to the published version of the manuscript.

**Funding:** This research was funded by the National Natural Science Foundation of China (21871275).

**Data Availability Statement:** The data presented in this study are available in this article.

**Acknowledgments:** D.D.S. is thankful to the CAS-TWAS President's Fellowship and G.A.S. is grateful to the Chinese Academy of Sciences for a President's International Fellowship for Visiting Scientists.

**Conflicts of Interest:** The authors declare no conflict of interest.

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
