# Peer review of "LLDPE-like Polymers Accessible via Ethylene Homopolymerization Using Nitro-Appended 2-(Arylimino)pyridine-nickel Catalysts"

_catalysts, doi:10.3390/catal12090961_

Round 1

Reviewer 1 Report

Review on manuscript catalysts-1881423

This article describes synthesis and characterization of 4 new Ni complexes bearing para-nitro substituted 2-(arylimino)pyridine ligands and one meta-nitro substituted Ni complex. They were further studies as catalytic precursors for ethylene polymerization that were activated mainly with Et2AlCl and MMAO. It was found that these catalytic systems are able to produce uniformly branched PE with long side chains that resemble the structure of commercial LLDPE. The catalytic performance of synthesized complexes has been studied in detail as well as the properties of obtained PE samples. Authors also extensively compared new catalytic systems with reported ones provided detailed analysis of the polymer microstructure for selected samples.

The manuscript is well written and concise, providing new structure properties comparison for Ni-based polymerization catalysts and some relevant insights. The experimental work is extensive and provides enough novelty and impact to be published in Catalysts. The main results such as high thermal stability of 2-(arylimino)pyridine NI complexes and the thorough analysis of branched polyethylene microstructure are of interest to the scientists involved in the fields of olefin polymerization.

I would recommend the publication of this manuscript in Catalysts after addressing some major and minor issues.

Major issues

1). I would strongly suggest adding all pictures of the NMR and IR spectra of ligands L1-L5 and initial anilines A1-A5 (since A1-A3 where already available in the lab, alternatively the link can be added to the publication where their characterization is described), to supplementary information. It will provide more clarity and consistency to the presented characterization data in terms of purity of the compounds. It will also allow to compare the characterization of the compounds more easily for the researchers that may want to repeat the synthesis in the future.

The characterization of A4, prepared according to literature procedures, is also missing in the supplementary information.

2). I would suggest performing mass spectrum characterization of complexes, if possible, to enhance the characterization of powders. The X-ray data obtained using monocrystals can differ from what is observed in powdered samples after their crystallization. The fragmentation patterns on mass spectra can also provide valuable information regarding, for example, monomeric vs dimeric structure, since both will give the same results in elemental analysis. So, it may be incorrect to confirm monomeric composition using only elemental analysis (p. 9 l.343-345).

3). It would be nice to provide the pictures of DSC analysis for samples with dual melt temperatures (and optionally for other PE samples) in the supplementary information (at least representative ones).

4). The degree of crystallinity of PE samples may also be another interesting feature to discuss, especially for those that have dual melt temperatures. In the absence of pictures of DSC data it is hard to judge, but since these samples may also have amorphous phase (due to the branched structure) in addition to two types of crystals that are governing the presence of melting temperatures as proposed by the authors. Were the Tg transitions for amorphous phase present in any of the DSC analysis results?

5). Do authors have any suggestions regarding why the reported complexes provide such high Mw/Mn rations compared to structurally similar analogs previously reported in literature? Is it caused by introduction of NO2 substituent or any other foreseeable factors?

Minor issues

1). Could authors provide some explanation/suggestion why Mw/Mn values also decrease along with Mw when Ni/Al ration increased (Table 3, runs 1-6)?

2). The quality of Figures 2-9 seems low, compared to other figures in the presented PDF.

p.3 l.124 Since the toluene is used as a solvent, I suppose it should be “homogeneous solution” instead of “homogeneous melt”.

p.7 l.272 I suppose it should be (DME)NiBr2 instead of (DME)NiBr2·6H2O as in experimental part.

p.10 l.358 modified methylaluminoxane (MMAO: containing 20-25% Al(i-Bu)3) – this information should also be presented in Materials section of Experimental part

p.10 l.359 EASC should instead of ESAC

Author Response

Comments to the author: This article describes synthesis and characterization of 4 new Ni complexes bearing para-nitro substituted 2-(arylimino)pyridine ligands and one meta-nitro substituted Ni complex. They were further studies as catalytic precursors for ethylene polymerization that were activated mainly with Et2AlCl and MMAO. It was found that these catalytic systems are able to produce uniformly branched PE with long side chains that resemble the structure of commercial LLDPE. The catalytic performance of synthesized complexes has been studied in detail as well as the properties of obtained PE samples. Authors also extensively compared new catalytic systems with reported ones provided detailed analysis of the polymer microstructure for selected samples. The manuscript is well written and concise, providing new structure properties comparison for Ni-based polymerization catalysts and some relevant insights. The experimental work is extensive and provides enough novelty and impact to be published in Catalysts. The main results such as high thermal stability of 2-(arylimino)pyridine NI complexes and the thorough analysis of branched polyethylene microstructure are of interest to the scientists involved in the fields of olefin polymerization. I would recommend the publication of this manuscript in Catalysts after addressing some major and minor issues.

Author response: We thank the reviewer for their appreciation of the results and positive recommendation. We have carefully studied both the major and minor issues and made the necessary revisions.

Major issues

Point 1). I would strongly suggest adding all pictures of the NMR and IR spectra of ligands L1-L5 and initial anilines A1-A5 (since A1-A3 where already available in the lab, alternatively the link can be added to the publication where their characterization is described), to supplementary information. It will provide more clarity and consistency to the presented characterization data in terms of purity of the compounds. It will also allow to compare the characterization of the compounds more easily for the researchers that may want to repeat the synthesis in the future. The characterization of A4, prepared according to literature procedures, is also missing in the supplementary information.

Author response: We thank the reviewer for their useful suggestion regarding the anilines. Accordingly, we have now provided the synthetic procedures for anilines A3-A5 (Scheme S1 and S2) along with their NMR data. In addition, their NMR spectra are presented in the supplementary information. Since pure A1 and A2 were commercially available (A1: CAS number #163704-72-1 and A2: CAS number # PH008328), they are used directly. Finally, the complete characterization data for A3 and A4 can be found in references S3 and S4 in the supplementary information.

Point 2. I would suggest performing mass spectrum characterization of complexes, if possible, to enhance the characterization of powders. The X-ray data obtained using monocrystals can differ from what is observed in powdered samples after their crystallization. The fragmentation patterns on mass spectra can also provide valuable information regarding, for example, monomeric vs dimeric structure, since both will give the same results in elemental analysis. So, it may be incorrect to confirm monomeric composition using only elemental analysis (p. 9 l.343-345)

Author response: We thank the reviewer for their kind suggestion regarding mass spectra. Although the compositions of three representative precatalysts were identified using single X-ray diffraction as adopting monomeric (Ni3) or dimeric (Ni4 and Ni5) structures, monomeric active species are commonly considered to be operational in the ethylene polymerization. Consequently, we consider the elemental analysis of the precatalysts to be sufficient to confirm the purity of these nickel complexes.

Point 3. It would be nice to provide the pictures of DSC analysis for samples with dual melt temperatures (and optionally for other PE samples) in the supplementary information (at least representative ones).

Author response: We thank the reviewer for their comment regarding the DSC analysis. Accordingly, we have added the DSC thermogram of the PE sample generated using Ni4/MMAO (run 4, Table 4) and listed it as Figure S6 in the supplementary information.

Point 4. The degree of crystallinity of PE samples may also be another interesting feature to discuss, especially for those that have dual melt temperatures. In the absence of pictures of DSC data it is hard to judge, but since these samples may also have amorphous phase (due to the branched structure) in addition to two types of crystals that are governing the presence of melting temperatures as proposed by the authors. Were the Tg transitions for amorphous phase present in any of the DSC analysis results?

Author response: We thank the reviewer for their comments regarding the PE crystallinity. Accordingly, we have now provided the DSC thermogram of one polymer sample displaying dual melt temperatures (prepared using Ni4/MMAO) in the supplementary information (Figure S6). During our DSC analysis, the instrument was set to operate with the range of temperature from -20 oC to 160 oC. Since our polymer samples are low-density PEs, their Tg transitions are expected to appear around -100 oC. Therefore, within this range, -20 oC to 160 oC, there were no Tg transitions observed.

Point 5. Do authors have any suggestions regarding why the reported complexes provide such high Mw/Mn rations compared to structurally similar analogs previously reported in literature? Is it caused by introduction of NO2 substituent or any other foreseeable factors?

Author response: We thank the reviewer for their insightful comment. The introduction of a nitro group is considered as one of the factors contributing to the increase in the Mw/Mn ratio as its strong electron-withdrawing properties increase the Lewis acidic character of the cationic active species leading to stable chain propagation reaction. In addition, we believe that when EtAlCl2 was used as an activator, the generation of stable multi-site nickel species occurred resulting in high Mw/Mn values.

Minor issues

Point 1. Could authors provide some explanation/suggestion why Mw/Mn values also decrease along with Mw when Ni/Al ration increased (Table 3, runs 1-6)?

Author response: We thank the reviewer for their comment. We believe that when the Al:Ni molar ratio gradually increased, the chain transfer process from metal center to aluminum co-catalyst become more important over chain propagation leading to the decrease in the Mw and the Mw/Mn values.

Point 2. The quality of Figures 2-9 seems low, compared to other figures in the presented PDF.
Author response: Accordingly, we have improved the quality of Figures 2-9 in the revised manuscript.

Point 3.  p.3 l.124 Since the toluene is used as a solvent, I suppose it should be “homogeneous solution” instead of “homogeneous melt”.

Author response: Accordingly, we have corrected “homogeneous melt” to be “homogeneous solution”.

Point 4. p.7 l.272 I suppose it should be (DME)NiBr2 instead of (DME)NiBr2·6H2O as in experimental part.

Author response: Accordingly, we have corrected “(DME)NiBr2·6H2O” to be “(DME)NiBr2”.

Point 5. p.10 l.358 modified methylaluminoxane (MMAO: containing 20-25% Al(i-Bu)3) – this information should also be presented in Materials section of Experimental part

Author response: Accordingly, we have added the information: ‘modified methylaluminoxane (MMAO: containing 20-25% Al(i-Bu)3)’ into Experimental section.

Point 6.  p.10 l.359 EASC should instead of ESAC

Author response: Accordingly, we have corrected “ESAC” for “EASC”.

Reviewer 2 Report

Good piece of work. All the experiments and characterizations are thorough and properly done.

Narration of the article cab improved. Use of complex sentences makes it hard for the readers. A quick revision on the language would be appreciated (It will also help to filter the type errors e.g line no. 644 – previously reports - change as - previous reports).

Performances of other important nickel-based olefin polymerization catalysts are need to be discussed to show where exactly the current system stands. For example.  https://doi.org/10.3390/polym10010041; https://doi.org/10.1038/s41467-019-14211-0; https://doi.org/10.1002/anie.202207363 etc…

Determination of parameters such as ‘rate of monomer insertion’ and ‘rate of chain termination’ from the obtained analytical data and using these parameters to compare the reactivities of the catalytic systems can help to understand these systems better. Refer to : https://doi.org/10.1021/om500345r; https://doi.org/10.1021/om4006998.

Author Response

Comments and Suggestions for Authors: Good piece of work. All the experiments and characterizations are thorough and properly done. Narration of the article cab improved. Use of complex sentences makes it hard for the readers. A quick revision on the language would be appreciated (It will also help to filter the type errors e.g line no. 644 – previously reports - change as - previous reports).

Author response: We thank the reviewer for their kind comments. Accordingly, we have improved the ‘narration’ of the article and made sentences easier to understand for the readers. In addition, we have corrected “previously reports” as “previous reports” as well as other typographical errors.

Point 1. Performances of other important nickel-based olefin polymerization catalysts are need to be discussed to show where exactly the current system stands. 

For example.  https://doi.org/10.3390/polym10010041; https://doi.org/10.1038/s41467-019-14211-0; https://doi.org/10.1002/anie.202207363 etc…

Author response: We are grateful to the reviewer for their useful suggestions as to other reports to include. Consequently, we have carefully studied these papers and as a result included these articles on nickel precatalysts; these have been cited in our manuscript as references 12-14. In addition, a sentence has been added into the introductory text to cover these reports: ‘Elsewhere, N,N- and N,O-based nickel precatalysts have more recently been reported to demonstrate an aptitude to exhibit high activity and good thermal stability for ethylene homo-/copolymerization [13,14]’. Furthermore, we have also made some comparison with some very recently reported nickel complexes based on N,N-diaryl-α-diimine ligands (references 38-42 and 57) that show some similarity to the recommended references.

Point 2. Determination of parameters such as ‘rate of monomer insertion’ and ‘rate of chain termination’ from the obtained analytical data and using these parameters to compare the reactivities of the catalytic systems can help to understand these systems better. Refer to: https://doi.org/10.1021/om500345r; https://doi.org/10.1021/om4006998.

Author response: We thank the reviewer for their useful suggestions. Accordingly, we have calculated both the ‘rate of monomer insertion’ (Ri) and the ‘rate of chain termination’ (Rt) values and moreover provided some description/discussion in the main text regarding this data. In addition, the calculated values are presented in Table 3 and Table 4, while the recommended references are cited as references 43 and 44.